# Benchmark seasonal prediction skill estimates based on regional indices

John E. Walsh[1], J. Scott Stewart[2], Florence Fetterer[2]

[1]Alaska Center for Climate Assessment and Policy, University of Alaska, Fairbanks, AK 99709 USA
[2]National Snow and Ice Data Center, University of Colorado, Boulder, CO 80303 USA

*Correspondence to*: John E. Walsh (jewalsh@alaska.edu)

**Abstract.** Basic statistical metrics such as autocorrelations and across-region lag correlations of sea ice variations provide benchmarks for the assessments of forecast skill achieved by other methods such as more sophisticated statistical formulations, numerical models, and heuristic approaches. In this study we use observational data to evaluate the contribution of the trend to the skill of persistence-based statistical forecasts of monthly and seasonal ice extent on the pan-Arctic and regional scales. We focus on the Beaufort Sea where the Barnett Severity Index provides a metric of historical variations in ice conditions over the summer shipping season. The variance about the trend line differs little among various methods of detrending (piecewise linear, quadratic, cubic, exponential). Application of the piecewise linear trend calculation indicates an acceleration of the winter and summer trends during the 1990s. Persistence-based statistical forecasts of the Barnett Severity Index as well as September pan-Arctic ice extent show significant statistical skill out to several seasons when the data include the trend. However, this apparent skill largely vanishes when the data are detrended. In only a few regions does September ice extent correlate significantly with antecedent ice anomalies in the same region more than two months earlier. The springtime "predictability barrier" in regional forecasts based on persistence of ice extent anomalies is not reduced by the inclusion of several decades of pre-satellite data. No region shows significant correlation with the detrended September pan-Arctic ice extent at lead times greater than a month or two; the concurrent correlations are strongest with the East Siberian Sea. The Beaufort Sea's ice extent as far back as July explains about 20% of the variance of the Barnett Severity Index, which is primarily a September metric. The Chukchi Sea is the only other region showing a significant association with the Barnett Severity Index, although only at a lead time of a month or two.

## 1 Introduction

One of the most widely monitored variables in the climate system is Arctic sea ice. By any measure, Arctic sea ice has decreased over the past few decades (Box et al., 2019). September sea ice extent during the past 5-10 years has been approximately 50% of the mean for the 1979-2000 period (AMAP, 2017). The recent decline is unprecedented in the satellite record, in the period of direct observations dating back to 1850 (Walsh et al., 2016), and in paleo reconstructions spanning more than 1400 years (Kinnard *et al*., 2011). The recent reduction of sea ice has been less in winter and spring than in summer and autumn, resulting in a sea ice cover that is largely seasonal (AMAP, 2017). The increasingly seasonal ice cover contrasts with the Arctic Ocean's predominantly multiyear ice pack of the pre-2000 decades. When compared to the reductions of the spatial extent of sea ice, the percentage reductions of ice volume and thickness are even larger. Ice thickness decreased by more than 50% from 1958-1976 to 2003-2008 (Kwok and Rothrock, 2009), and the percentage of the March ice cover made up of thicker multiyear ice (ice that has survived a summer melt season) decreased from 75% in the mid-1980s to 45% in 2011 (Maslanik *et al*., 2011). Laxon et al. (2013) indicate a decrease of 64% in autumn sea ice volume from 2003-08 to 2012. The portion of the Arctic sea ice cover comprised of older thicker ice has decreased from 45% in 1985 to 21% in 2017 (NOAA, 2018).

While the loss of sea ice is generally presented in terms of pan-Arctic metrics, regional trends can be quite different from the pan-Arctic trends. The Bering Sea, for example, showed a positive trend of coverage (fewer open water days) from 1979 through 2012 (Parkinson, 2014), However, the positive trend of Bering Sea ice largely vanishes when the most recent winters (especially 2017-18) are included. By contrast, the Chukchi and Beaufort Seas to the north of the Bering Sea have shown some of the largest decreases of summer ice coverage in the entire Arctic (Onarheim et al., 2018). Another area of strong decrease of ice coverage has been the Barents/Kara Sea region.

The Beaufort Sea serves as an illustrative example of the impacts of trends and variability of sea ice. The number of open water days immediately offshore of the Beaufort coast has been 60-120 in recent years. Parkinson's (2014) Figure 2 shows that the number of open water days increased by 20-30 days per decade over the period 1979-2013. However, as recently as the 1970s, there were summers with little or no open water in this region, as described by Crowley Maritime, one of the major barge operators in the Alaska region:

"With pipeline construction well underway in 1975, the Crowley summer sealift
flotilla to the North Slope faced the worst Arctic ice conditions of the century. In fleet
size, it was the largest sealift in the project's history with 47 vessels amassed to carry
154,420 tons of cargo, including 179 modules reaching as tall as nine stories and
weighing up to 1,300 tons each. Vessels stood by for nearly two months waiting for the
ice to retreat. Finally in late September the ice floe moved back and Crowley's tugs and
barges lined up for the slow and arduous haul to Prudhoe Bay. When the ice closed again,
it took as many as four tugs to push the barges, one at a time, through the ice".

— *From Crowley Maritime, 50 Years of Service in Alaska (2002)*

As will be shown, the contrast between present-day ice conditions and the Crowley
experience of the 1970s is largely a manifestation of the trend of Beaufort Sea ice cover.
However, sea ice also exhibits large year-to-year variability, which has been superimposed on
the recent trend towards less sea ice in the Arctic. This variability challenges users of coastal
waters in various sectors and lies at the heart of the sea ice prediction problem.  While the
climatological seasonal cycle and even observed trends provide an initial expectation for the sea
ice conditions that will be present in a particular region at a particular time of year, the
departures from the climatological mean, whether or not the mean is adjusted for a trend, is
affected by the atmospheric forcing (winds, air temperatures, radiative fluxes) and oceanic
forcing (currents, water temperatures) of the particular year in addition to antecedent ice
conditions themselves. These departures have a large component of internal variability and hence
are difficult to predict over monthly and seasonal timescales (Serreze et al., 2016), raising
questions about the extent to which sea ice variations may be predictable.
Fully coupled models, which determine both the atmospheric and ocean/ice conditions
prognostically, are now used increasing often for seasonal sea ice predictions.  Ensembles of
coupled simulations are generally run because of the chaotic nature of the climate system. These
models can be run for much longer time periods than the observational sea ice record, so they
can provide statistics of sea ice persistence (autocorrelations) and other atmosphere-ocean-sea
ice relationships subject to the "perfect model" assumption, whereby model output is treated as if
it were data from the real world.  In other words, the model's world is regarded as equivalent to
the actual climate system.  Examples of studies employing the "perfect model" approach are
Holland et al. (2011), Blanchard-Wrigglesworth et al. (2011), Day et al. (2014), Bushuk et al.
(2017) and Bushuk et al. (2018).  In these model simulations, autocorrelation of sea ice
anomalies tends to be greater in the model results than in observational data (e.g., Blanchard-
Wrigglesworth et al., 2011, their Fig. 2; Day et al., 2014, their Fig. 1). However, skill in perfect
model simulations is not due solely to persistence, as physical and dynamical processes driving
changes in sea ice can be captured by models.
The skill of persistence-based statistical forecasts of sea ice variations beyond the mean
seasonal cycle and on-going trends is the main focus of this paper.  While various prior studies
(cf. Section 2) have utilized broader approaches to evaluating sea ice predictability and the skill
of forecasts, the present study is limited specifically to statistical predictions of regional (and
pan-Arctic) September sea ice extent based on auto-correlation (anomaly persistence, often
referred to as "memory") and lagged cross-correlations with other sea ice coverage quantities.
Other approaches to sea ice predictability include the use of models, which can be initialized to
obtain deterministic forecasts verifiable with observations or which can be run for long periods
in a coupled mode to assess predictability of sea ice within the "model's world" (irrespective of
observations).  We also do not use atmospheric or oceanic predictors in our evaluation of
persistence-based predictability.  Atmospheric predictors in the form of known teleconnection
patterns have been used in statistical studies by Drobot (2003) and Lindsay et al. (2008), while
Bushuk et al. (2017) concluded from a dynamical model hindcast study that ocean temperature
initialization contributes to skill of seasonal forecasts of sea ice in the North Atlantic subarctic
seas.  This conclusion is consistent with Blanchard-Wrigglesworth et al.'s (2011) finding that
across-season persistence of ocean temperature anomalies makes a detectable contribution to
seasonal sea ice predictability. The present study also does not include ice thickness, which has
been shown to be an important source of predictive skill for summer sea ice (Day et al. 2014;
Collow et al. 2015; Dirkson et al. 2017; Zhang et al. 2018).  Guemas et al. (2016) provide a
review of the various approaches to sea ice prediction and sources of predictability.
The present paper extends the temporal window of Drobot's (2003) study of the
predictability of Beaufort-Chukchi sea ice.  Drobot used data from 1979-2000 to assess
predictability of a measure of Beaufort Sea summer ice severity (Section 3 below) based on
antecedent sea ice conditions as well as several atmospheric indices. While the present study will
not include the type of multiple-predictor evaluation carried out by Drobot, it will provide a more
comprehensive and updated assessment of sea ice anomaly persistence in a predictive context.
Drobot (2003) found that, in predictions based on indicators from the previous seasons, the
limited sample of years used in developing the statistical models raises questions about broader
applicability. In this regard, Drobot (2003, p. 1161) states "…if the Arctic climate changes, the
methods described here will need to be altered".  In fact, the Arctic climate and, in particular, its
sea ice regime, have changed with the unprecedented retreat of sea ice in the post-2000 period.
The impact of the trend on statistical predictability is a focus of the present paper. We note,
however, that evolving physical relationships that underlie trend-related changes in statistical
relationships are not addressed in the present study.
In the present paper, we use the autocorrelation statistic to quantify the skill of persistence as
a control forecast of pan-Arctic and regional sea ice extent.  In addition to utilizing the more
conventional metric of ice extent in regional and pan-Arctic domains, we include a regional sea
ice index developed in the 1970s to capture interannual variations of marine access in the
Beaufort Sea.  A primary focus of the evaluation is the method of detrending the data, as various
alternative methods have not been fully explored in the literature. We show that the piecewise
linear method generally results in the smallest residual variance about the trend line, and we then
perform an across-region synthesis of information on the break-points of the two-piece linear
trend lines in different seasons.  Our period of analysis extends back to 1953, which results in a
considerably larger sample of years than the more commonly used satellite period (1979
onward).  Finally, we examine lagged cross-correlations to determine whether pan-Arctic ice
extent or Beaufort Sea summer ice conditions are foreshadowed in a statistical sense by
antecedent ice conditions in particular subregions of the Arctic.
More generally, the results presented here can serve to provide a baseline for distinguishing
contributions to seasonal sea ice forecast skill arising from climatological sea ice coverage, sea
ice persistence, and sea ice trend.  This baseline can, in turn, serve as benchmarks for measuring
improvements achieved by more sophisticated prediction approaches such as dynamical models,
analog systems, neural networks and other more comprehensive statistical methods. The Sea Ice
Outlook, coordinated by the Sea Ice Prediction Network now in its Phase 2
(https://www.arcus.org/sipn/sea-ice-outlook, accessed 11 February 2019), provides an annual
compilation of seasonal sea ice forecasts, which are grouped into three categories:
physical/dynamical models, statistical methods, and heuristic approaches. While the
methodology used in this paper falls into the statistical category, the distinctions between (a)
pan-Arctic and regional skill and (b) trend-derived and interannual forecast skill are relevant to
all three approaches to sea ice prediction.

**2.  Previous work**
Baselines for persistence-based predictions have been established in previous studies
(e.g., Blanchard-Wrigglesworth et al., 2011; Day et al., 2014; Bushuk et al., 2017, 2018). While
these studies generally used long control runs from climate models, their observational records
were limited to the post-1979 period of satellite data.  The present study is based on a longer
observational record (back to 1953 rather than 1979).  The main intent of the paper is to show
how detrending is a key step in the depiction of persistence-based statistical predictions.  We
illustrate the effect of detrending for both pan-Arctic ice extent and regional metrics in order to
show that predictive applications on both scales must address detrrending in a rigorous way, and
that there are various alternatives for detrending.  While these alternative detrending strategies
are known, the relative effectiveness of the various alternatives has not been addressed in
previous studies.  Goldstein et al. (2016; 2018) come closest by comparing representations based
on linear trends and discontinuities in the mean. An additional novel outcome of the present
study is the synthesis of break-point information.
The extension back to 1953 is especially noteworthy because of the recent reduction of
Arctic sea ice coverage has occurred almost entirely in the post-1978 period of satellite coverage.
On both pan-Arctic and regional scales, ice extent was relatively stable during the 1950s, 1960s
and 1970s, although interannual variations were then, too, a prominent feature of the time series
(Walsh et al., 2016).  While Drobot (2003) and Lindsay et al. (2008) made use of sea ice data
extending back to the 1950s, there has been no systematic comparison of sea ice anomaly
persistence during the satellite era with anomaly persistence over longer time periods.
**3. Metrics of sea ice coverage**
Historical variations of sea ice are documented using various metrics, including pan-Arctic
sea ice extent, ice-covered area, and thickness.  Sea ice extent is the total area within the ice
edge, which is typically taken to be the 15% contour of sea ice concentration.  Ice extent is
readily obtainable from satellite measurements, as is the actual ice-covered area if the open water
within the ice edge is accurately depicted.  Surface-based observations from ships or coastal
locations typically capture only the ice edge and are therefore useful primarily in the mapping of
ice extent.  While digitized records of ice extent reaching back to the 1800s exist, there are no
such historical products for ice thickness.  *In situ* measurements of ice thickness are sparse in
space and time, as are submarine sonar measurements, which are not only sparse but often
remain unavailable.  Holt (2018) provides a rare compilation if *in situ* measurements. Satellite-
derived estimates of ice thickness are subject to considerable uncertainty and have only recently
come into use (e.g., Lindsay and Schweiger 2015), while dynamic-thermodynamic model-based
reconstructions of historical sea ice thickness variations have only recently been attempted
(Schweiger et al., 2019).

To explore the statistical skill that may be inherent in the spatial distribution of sea ice, we

compute ice extent using the gridded Arctic-wide sea ice concentration product known as
"Gridded Monthly Sea Ice Extent and Concentration, 1850 Onward (Walsh et al., 2015), referred
to in the National Snow and Ice Data Center (NSIDC) catalog as G10010.  This dataset is based
on observations from approximately 15 historical sources between 1850 and 1978: the earliest
are whaling records, and the most complete, in terms of coverage, are the Arctic-wide analyses
that what is now the U.S. National Ice Center began in the early 1970s.  Beginning in 1979, sea
ice concentrations from passive microwave data are used exclusively in G10010. Ice
concentration fields on the 15$^{th}$ of each month were taken from the *NOAA/NSIDC Climate Data*
*Record of Passive Microwave Sea Ice Concentration, Version 2 (Meier et al., 2013).*

Prior to the 1950s, most observations were from near or just within the ice edge.  If only the

ice edge position was known, a gradient of ice concentration within the edge was imposed in
order to integrate the observations into G10010.  The gradient was based on a climatology
constructed from the passive microwave data.  Spatial and temporal gaps in observations were
filled using an analog technique that is described in the data product documentation.  Each
month's sea ice concentration field in G10010 is an estimate of conditions at one time in the
month, nominally the 15$^{th}$ day of the month (or as close to the 15$^{th}$ as data were available). The
fields are at quarter-degree resolution. From these fields one can derive monthly sea ice extent
values.  Sea ice extent is computed as the area, in sq km, covered by all cells that contain ice in
any concentration greater than 15%.  Sea ice extent is always greater than or equal to the actual
ice-covered area, which excludes the area of open water within the main ice pack.
Various studies (e.g. Partington et al., 2003; Agnew and Howell 2002) have shown that
passive microwave-derived sea ice data tend to underestimate ice concentration when compared
with operational analyses.  The *Climate Data Record of Passive Microwave Sea Ice*
*Concentration* is a blend of output from two algorithms that results in higher ice concentrations
overall for a better match with the operational analyses that predate the satellite record.   Even
so, one might expect to see a discontinuity in the G10010 time series of ice extent when the
passive microwave record starts in 1979, but this is not evident (see Fig. 10 in Walsh et al.,
2016).  While G10010 gives a record of ice extent that has realistic variability back to 1850, it is
difficult to assign an uncertainty to the concentration fields and ice extent values derived from
them. Ice extent will be more accurate than actual ice-covered area because there are many more
observations of the ice edge than of the concentrations within interior pack.  For this reason, we
base our analysis on ice extent.  It should be noted, however, that persistence time-scales of pan-
Arctic sea ice area have been shown in previous studies (e.g., Blanchard-Wrigglesworth et al.,
2011) to be longer than those of pan-Arctic sea ice extent because high-frequency forcing can
change ice extent more than it changes ice area (i.e., by converging or diverging ice floes in the
absence of ridging or melt).
G10010 was used to compute the time series of monthly sea ice extent for the pan-Arctic
domain and various Arctic subregions in which sea ice is at least a seasonal feature. The
regionalization adopted here follows that of the MASIE (Multisensor Analyzed Sea Ice Extent)
product available from the National Snow and Ice Data Center
(http://nsidc.org/data/masie/browse_regions, accessed 27 Dec 2018).  MASIE (NIC and NSIDC,
2010) regions are defined by the U.S. Navy-NOAA-Coast Guard National Ice Center (NIC) on
the basis of operational analyses areas.  We use the following MASIE regions: (1) Beaufort Sea,
(2) Chukchi Sea, (3) East Siberian Sea, (4) Laptev Sea, (5) Kara Sea, (6) Barents Sea, (7) East
Greenland Sea, (8) Baffin Bay/Davis Strait, (9) Canadian Archipelago, (10) Hudson Bay, (11)
central Arctic Ocean and (12) Bering Sea.  There are several other MASIE regions (Baltic Sea,
Yellow Sea, Cook Inlet) that are not used here because they are not geographically connected
with the main Arctic sea ice cover. Figure 1 shows the regions.
We also make use of the long ice extent record provided by G10010 to investigate the extent
to which the Barnett Severity Index, or BSI, may be statistically predictable from antecedent ice
extent. The BSI is directly relevant to offshore navigation applications in the Beaufort Sea. It is
a metric of the severity of ice conditions, such as conditions encountered by barges resupplying
the North Slope. The BSI is determined once per year, at the end of the summer shipping season,
by analysts at the NIC. It is a unit-less linear combination of five parameters: 1) the distance in
nautical miles from Point Barrow northward to the ice edge on 15 September, 2) the distance
from Point Barrow northward to the 4/8th ice concentration line on 15 September, 3) the number
of days the entire sea route from the Bering Strait to Prudhoe Bay is ice-free in a calendar year,
4) the number of days the entire sea route to Prudhoe Bay is less than or equal to 4/8th ice
concentration in a calendar year, and 5) the temporal length of the navigable season, defined as
the time period from the initial date the entire sea route is less than 4/8th ice concentration to 1
October (Barnett, 1980). Figure 2 is a time series of the BSI reconstructed from gridded sea ice
concentration data (see Appendix). Higher values indicate less severe ice conditions.
**4  Methods**
As shown in Figure 3, Arctic sea ice extents have generally been decreasing over the post-
1953 period of this study. The Beaufort Sea is a prime example of a region in which summer
and autumn sea ice coverage has been decreasing, although winter (March) sea ice extent in the
Beaufort Sea shows no trend or variability because the ice edge extends to the coastline in March
of every year, essentially eliminating year-to-year variations. Consistent with the September
decrease of Beaufort ice extent, the BSI has been increasing over the past few decades (Figure
2). Two time series containing trends over time can show a correlation simply because the trends
are present in the time series. A trend can be used as a predictive tool by assuming its
continuation into the future. However, a trend can inflate persistence-based forecast skill when a
variable is used to predict itself (assuming the historical trend continues into the future). Indeed,
depictions of time-variations of a quantity such as sea ice extent are often shown as departures
from a trend line in order to highlight the interannual variations. One of our main interests in this
study is whether or not interannual variations of preceding regional ice extents correlate with
later BSI values. In order to exclude the effect of the overall trends in the correlation of these
time series, we detrend the data and explore various methods for doing so.
The choice of a function with which to de-trend the time series should be determined by
features of the series itself.  The detrended time series should exclude the general tendency to
change over time, but preserve a measure of the year-to-year variability of the series. The
previous studies cited in Section 2 (e.g., Blanchard-Wrigglesworth et al., 2011; Sigmund et al.,
2013; Day et al., 2014; Bushuk et al., 2017, 2017) have generally relied on least-squares linear
fits, while Dirkson et al. (2017) suggested the use of a quadratic fit for detrending pan-Arctic sea
ice area.. Goldstein et al. (2016, 2018), by contrast, showed that discontinuous changes in the
mean better captured time series (such as open water area) characterized by abrupt changes. In
the spirit of the Goldstein et al. studies, we explore various options for detrending a time series
such as those in Figures 2 and 3, for which the changes are more pronounced in recent decades
than in earlier decades.  In such cases, a single multi-decadal trend line cannot be expected to
optimally represent the historical evolution.
We explored several functional forms that fit the time series, including linear, quadratic,
cubic, and exponential functions.   We found that a simple two-piece linear function – wherein
the data are modelled by two line segments that intersect at a 'break-point' year – had the lowest
average RMS difference between the time series and the fitted function, although fits using other
functions had only slightly larger RMS differences.  This choice of the detrending fit has the
additional feature of giving a sense of when the ice extent began to change more rapidly.
The two-piece linear fits were obtained by using standard statistical algorithms.  A
function defined by two intersecting half-lines can be specified by the coordinates of one point
on each half-line and the intersection point.  With the x-axis as time, and the y-axis as the value
of the sea ice extent, the x-values of the non-intersecting points can be chosen to be 1953 and
2013, the first and last years of the BSI dataset.  This leaves four values for the function to fit:
the series value in 1953, the series value in 2013, and the year and value at the intersection point,
also referred to here as the break-point.  We note that the break-point is not specified by the user
but is determined by the algorithm so that the fit to the time series is optimized. The "curve_fit"
method is defined in lines 504-794 of the file
https://github.com/scipy/scipy/blob/master/scipy/optimize/minpack.py  This method performs a
least-squares fit to the function by modifying the equation's parameters.  A starting "guess" of the
equation parameters is provided by the user.  The 'curve_fit' method of the SciPy numerical
library are then used to algorithmically modify the equation parameters to find the best two-piece
linear fit to the function.

In Figure 4, we show the piecewise linear fit together with quadratic, cubic and exponential

fits to the time series of the BSI and the September Beaufort Sea ice extent.  In the case of the
two-piece linear fit, the break-point -- found by the curve_fit procedure to best fit the data -- is in
the early 1990s for both sea ice metrics.  It is visually apparent from Figure 4 that all four fits are
comparable in terms of the overall magnitudes of the departures from the trend lines.  The root-
mean-square departures from the various trend lines indeed differed by less than 10%. Because
the two-piece linear fit was usually the best fit and also provided a clear estimate -- the break-
point year -- of when the recent rate of sea ice loss accelerated, we chose the two-piece linear for
the remainder of this study.  The break-points are computed separately for each region allowing
the use of the two-piece linear fit to compare the timing of the change in ice loss rate among the
various subregions.

After using the 'curve_fit' method to find two-piece linear fits to the BSI and the regional

and pan-Arctic sea ice extent time series, we subtracted this linear fit from the original data to get
a detrended time series for each sequence.  We were then able to use the 'linregress' method
from the SciPy (Jones et al., 2001) software library to find the linear relationship between the
detrended regional monthly extent values and both the detrended BSI and the detrended pan-
Arctic extent.  We then used the 'stats' method from SciPy to compute the square of the Pearson
correlation coefficient ($R^2$) and the p-value estimate of statistical significance for this
relationship.

**5  Results**

As noted in Section 2, previous studies (Bushuk et al., 2018) have evaluated the persistence

of regional ice extent over the post-1978 period of satellite observations.  Here we extend this
evaluation to encompass a longer period dating back to 1953 in order to assess the stability of the
persistence statistics.  Specifically, for each region in Figure 1, we have correlated the September
ice extent with the ice extent of antecedent months for the 1953-2013 and 1979-2013 periods.
Figure 5 compares these persistence values (autocorrelations at multimonth lags), for the
antecedent months of March, May and July in a subset of regions. Because the regions chosen
were those that have interannually varying ice cover in September, regions such as the Bering
Sea, Hudson Bay, the Sea of Okhotsk and the Baltic Sea were excluded.  The correlations for the
non-detrended and detrended ice extents are shown in panels (a) and (b), respectively.
For most of the regions, the inclusion of the earlier decades does not have a notable impact
on the persistence from July to September, and detrending the data does not change this
conclusion. However, in the non-detrended results of Figure 5a, the March-to-September and
May-to-September correlations change substantially in a few regions.  The Baffin Bay March-to-
September correlations increase from 0.00 to 0.34 when the earlier decades are eliminated,
largely as a result of the post-1979 trend: the post-1979 correlation is statistically significant (p <
0.05) based on a Wald test with t-distribution of the test statistic and a two-sided p-value for a
null hypothesis that the slope is zero.  The pan-Arctic correlations, as Figure 5a shows, are
higher than the correlations for the individual regions, also increase when the earlier decades are
eliminated.  In the Greenland Sea, the correlations from March and May decrease substantially
and lose statistical significance when the earlier decades are eliminated.
As shown in Figure 5b, the detrending generally reduces the magnitudes of the
correlations between September and the  earlier months, both for the longer post-1953 periods
and the shorter post-1979 periods: The March-to-September correlations based on the detrended
data for the longer/shorter periods are: -0.05/0.20 for Baffin Bay, 0.20/0.13 for the Barents Sea,
0.00/0.00 for the Beaufort Sea (no March variance), 0..00/0/00 for the Canadian Archipelago (no
March variance), -0.15/0.00 for the Chukchi Sea, 0.07/0.21 for the East Siberian Sea, 0.25/-0.03
for the Greenland Sea, 0.03/0.03 for the Kara Sea, and 0.07/0.18 for the Laptev Sea.  The
corresponding 5% significant levels are 0.26/0.33.  Except for the Canadian Archipelago and
East Siberian regions, the May-to-September correlations in Figure 5b are also small (<0.40).
Only the July-to-September correlations are above 0.40 in all regions for the detrended data.  In
view of generally small magnitudes of the March-September and May-September correlations in
Figure 5b, we conclude that the springtime "predictability barrier" (Lindsay et al., 2008; Day et
al., 2014; Bushuk et al., 2018) in regional forecasts based on persistence of ice extent anomalies
is not reduced by the inclusion of several decades of pre-satellite data.
Because changes of trend have not been addressed systematically in previous evaluations of
Arctic sea ice trends, we synthesized the break-point information across all regions and calendar
months (January-September) included in our study.  The synthesis was limited to only those
regions and calendar months in which the two-piece linear fit reduced the root-mean-square
residual by at least 5% relative to the one-piece linear best fit.  Figure 6 groups the break-points
into five year periods ending in 1955, 1960,…, 2015.  In order to capture the seasonality of the
break-points, we present separate plots for (a) the entire January-September period, (b) January-
March (winter), (c) April-June (spring), and July-September (summer).  As shown in panel ((a),
nearly all the break-points occur in the second half of the study period, with a maximum in 1991-
1995.  The 1991-1995 period has the most break points of any 5-year period, and the 1990s have
nearly as many break points as all the other decades combined.  The small secondary peak in the
1960s represents eight break-points scattered across the regions and seasons (Cook Inlet in
January and February; East Siberian Sea in February and April; pan-Arctic in July; Hudson Bay
in August; Baffin-St. Lawrence and Bering in September), showing no systematic pattern that
would suggest a meaningful signal. The winter and summer seasons are the primary contributors
to the maximum in the 1990s, as the spring break points are evenly distributed through the latter
half of the study period.  However, spring has the fewest (12) break-points overall, while the
summer has the most (26). The break-points for our focal metrics, the BSI and September pan-
Arctic ice extent, are 1991 and 199, respectively, consistent with the distribution in Figure 6.
These two metrics are included in the results summarized in Figure 6.  One may conclude that
the 1990s, and to a lesser early 2000s, represent the shift to a more rapid rate of sea ice loss.  If
one is to argue for a "regime shift" in Arctic sea ice loss (Lenton, 2012), this period would be the
leading candidate.

In order to illustrate the effect of the detrending and to show which regions contribute the

most explained variance to pan-Arctic sea ice extent, Figure 7 shows the squares of the
correlations ($R^2$) between September pan-Arctic ice extent and the concurrent ice extent in each
of the subregions.  The figure shows values of $R^2$ before detrending (upper numbers, regular
font) and after detrending (lower numbers, bold font). With the trend included, the $R^2$ values are
relatively high in most regions (except for the Bering Sea), ranging from 0.32 to 0.71; the
corresponding correlations (R) range from 0.57 to 0.84. Based on the t-test described earlier,
these correlations all exceed the 95% significance thresholds, which range from 0.26 ($R^2 = 0.07$)
for a 60-year sample with no autocorrelation to 0.38 ($R^2 = 0.14$) for a 60-year sample with an
autocorrelation of 0.4.  None of the regional or pan-Arctic ice extent autocorrelations exceeded
0.40.  Because these correlations are dominated by the trend, the larger values appear in the
regions with trends that are most similar to the pan-Arctic trend.  When the data are detrended,
the explained variances are much smaller ($R^2$ values in bold font in Figure 7) although still larger
than the 95% significance thresholds for a 60-year sample (R = 0.26, $R^2$ = 0.07). These smaller
values indicate the relative contributions of regional variations to the interannual variations of
pan-Arctic ice extent.  According to Figure 7, the regions contributing most strongly to
September pan-Arctic sea ice variations (including trends) are the Beaufort, Chukchi and East
Siberian Seas.  After the data are detrended, the regions contributing most to September pan-
Arctic sea ice variations are the East Siberian and Laptev Seas.  The relatively large contribution
of the Laptev Sea is consistent with the "dynamical preconditioning" hypothesis of Williams et
al. (2016).  The variances of the detrended pan-Arctic September extents explained by the East
Siberian and Laptev Seas are indeed among the largest of all the regions, although the Chukchi
Sea's interannual variance is essentially as large.
Figure 8 shows the squares of the correlations between the annual BSI and regional
September ice extent before the detrending of both variables (top numbers) and after detrending
(bottom numbers).  While the actual correlations between the BSI and regional extent are
generally negative, the $R^2$ values plotted in Figure 8 are positive.  Large values of $R^2$ appear in
most regions when the trend is included (upper numbers) because the BSI has a strong positive
trend over time while September ice extent in most regions has a negative trend.  The $R^2$ values
are much weaker in regions away from the Beaufort Sea when the trends are removed (lower
numbers in Fig. 8). The detrended $R^2$ values show the spatial representativeness of the BSI as a
measure of interannual variations in each region. Figure 8 shows that the regions of significant
explained variance include the Canadian Archipelago to the east as well as the Chukchi Sea to
the west.  However, the "scale of influence", if measured by the area of significant correlation, is
smaller for the BSI in Fig. 8 than for pan-Arctic ice extent in Fig. 7.
Because the potential for seasonal predictions is a key motivation for this study, we
examine cross-correlations in which the predictands (pan-Arctic ice extent and the BSI) lag
potential predictors (regional ice extents) by intervals ranging from zero (no lag) to several
seasons.  Cross-correlations between non-detrended and detrended September pan-Arctic and
regional ice extents are summarized in Tables 1 and 2 respectively. Cross-correlations between
non-detrended and detrended BSI and regional ice extent are given in Tables S1 and S2
respectively. In all cases, the numerical values are the $R^2$ values.  In order to illustrate the
contribution of the trend to the apparent forecast skill, we present these correlations graphically
for the regions which show the strongest associations with the September predictands.  Figure 9
shows the $R^2$ values for cases in which September pan-Arctic ice extent lags by 0, 1, 2,…,8
months the ice extent in five subregions: the Beaufort, Chukchi, East Siberian, Barents and
Laptev Seas. The red bars correspond to correlations computed from the data with the trends
included. Not surprisingly, the $R^2$ values are largest at zero lag.  The rates at which the
correlations decrease with increasing lag vary regionally, reaching zero by 3-4 months for the
Beaufort, Chukchi, and East Siberian Seas. The zero-month lag values are quite large for the
Beaufort, Chukchi, and East Siberian regions, where they exceed $R^2 = 0.7$ (R = 0.84).

However, after detrending (using the two-piece linear best fits), most of the apparent forecast

skill is lost. As shown by the blue bars in Figure 9, nearly all the predictability from the Barents
and Chukchi Seas vanishes with the detrending, while only small fractions of explained variance
remain at non-zero lags when sea ice extents for the Beaufort and East Siberian Seas are the
predictors.  For example, when the regional extent leads by two months (July), the fractions of
explained variance are approximately 0.16 and 0.10 (R ~ 0.40 and 0.32) for the East Siberian and
Beaufort Seas, respectively.  The implication is that the persistence of interannual variations about
the trend line makes only small contributions to interannual variations of pan-Arctic sea ice extent,
and that these small contributions result mainly from the Pacific sector of the Arctic. As indicated
by Figure 9, the pan-Arctic extent of July and August correlates more highly than any regional
extent with September pan-Arctic ice extent in both the non-detrended and the detrended data (see
also Tables 1 and 2).
The lagged $R^2$ values relevant to predictions of the Barnett Severity Index are shown in
Figure 10.  Because the BSI is based primarily on ice conditions in the Beaufort Sea in August
and September, it is not surprising that the correlation is largest for the Beaufort's ice extent in
September, when the $R^2$ value is approximately 0.8 for data that are not detrended.  The August
and September values for the Chukchi are essentially as large as the corresponding Beaufort
values, indicating a spatial coherence of the variations (with trends included) in the two regions.
The antecedent extents in the East Siberian and Barents regions also explain statistically
significant fractions of the variance when the trends are included.
The blue bars in Figure 10 are the lagged $R^2$ values based on the detrended data. Because the
trend's contribution to the forecast skill has been removed, these correlations provide the most
meaningful assessment of the seasonal forecast skill if the BSI based on antecedent ice
conditions.  The largest correlations are for the Beaufort Sea, where the explained variances
decrease from about 0.55 (R ~ 0.74) in September to about 0.10 (R ~ 0.32) in June. The
correlations for the Chukchi are only slightly smaller, but the BSI variance explained by all other
regions is less than 10%.  The percentage of variance explained by the antecedent ice extent  of
the nearby regions (Beaufort, Chukchi, East Siberian Seas) is less than one might have
anticipated, given that the BSI includes information on the length of the navigation season,
which can begin well before September, i.e., as early as July in some years.

While the results presented here imply that the persistence of detrended sea ice anomalies

provide only limited forecast skill, a key question is: How much better are sea ice forecasts based
on other approaches? The Sea Ice Prediction Network's Sea Ice Outlook (SIO) consists of
seasonal forecasts of September sea extent based on a variety of approaches (numerical
modelling, statistical and heuristic) on an annual basis beginning in 2008.  In most years, several
dozen individual (or groups) provide the SIO with September sea ice extent predictions based
data and other information available at the end of May, June and July.  A compilation of SIO
results from 2008-2018 enables a quantitative comparison of the skill of the SIO and persistence-
based forecasts. (In this case, persistence was evaluated from the mean ice extents in the
National Snow and Ice Data Center's G02135_v3.0:
ftp://sidads.colorado.edu/DATASETS/NOAA/G02135/seaice_analysis/, accessed 27 Dec 2018).
The median absolute error of the all-forecaster average SIO issued in July of 2008-2018 is 0.32
million $km^2$, while the corresponding median absolute error of forecasts of persistence of the
departure from the trend line of the pan-Arctic ice extents of May, June, July and August are
0.43, 0.22, 0.25, and 0.09 million $km^2$. Thus the SIO forecasts issued in July outperform the
trend-line anomaly persistence forecasts from May, but not from June, July or August.
Persistence of the previous September's deviation from the trend line has a median absolute 0.37
million $km^2$, while simple persistence of the previous year's actual value has an error of 0.40
million $km^2$.  The corresponding root-mean-square errors (in millions $km^2$) are 0.57 for SIPN;
0.67, 0.46. 0.42, and 0.18 for persistence of the trend-line departures of May, June, July and
August; 0.68 for persistence of the trend-line departure from the previous September; and 0.67
for persistence of the actual extent from the preceding September.  The SIO forecasts used in this
comparison were averages of all forecasts submitted to SIO, so it is quite possible that some
individual forecasters participating in the SIO perform considerably better.  Nevertheless, it is
apparent that sea ice anomaly persistence is a challenging control forecast and a respectable
competitor for forecasts issued by the scientific community

**6 Conclusion**

The substantial decrease of Arctic sea ice over the past several decades is well documented

(Cavalieri and Parkinson, 2012; Parkinson, 2014; Onarheim et al., 2018). Of all the regions
considered here, only the Bering Sea does not show a negative trend, and the Bering trend is
positive only in January-April (Onarheim et al., 2018, their Table 1). Moreover, the extreme
minima of Bering Sea ice during the past two winters (2016-17 and 2017-18) are starting to bring
the Bering's trend into alignment with the other regions of the Arctic.

The prominence of the trends in the time series of regional as well as pan-Arctic ice extent

makes it important to distinguish the contribution of the trend from other sources of forecast
skill. In this study we explored the use of several methods of detrending in order to evaluate the
use of ice anomaly persistence (autocorrelation) and regional cross-correlations as predictors of
ice variations. The two-piece linear trend evaluations generally have break-points in the 1990s,
indicating that the rate of ice loss has been greater in the past two decades than in the earlier
portion of the satellite era that began in 1979.

Based on the raw (not detrended) time series, the antecedent ice extents in a substantial

fraction of the Arctic regional seas explain statistically significant fractions of variance of
September pan-Arctic ice extent and also of the Barnett Severity Index, which is more specific to
the Beaufort Sea. Statistically significant portions of variance of both September metrics are
explained by the regional ice extents of prior seasons. However, this predictive "skill" is
attributable primarily to the trends in the data. Removal of the trend leaves little forecast skill
beyond a month or two when the forecast method is limited to the relatively simple statistical
correlations utilized here. The low persistence-derived skill for the detrended September pan-
Arctic ice extent is consistent with the findings of Stroeve et al. (2014) based on the Sea Ice
Outlook as part of the Study of Environmental Arctic Change (SEARCH). Moreover, our
inclusion of data back to the early 1950s shows that the springtime "predictability barrier" in
regional forecasts based on persistence of ice extent anomalies is not reduced by the inclusion of
several decades of pre-satellite data.

It must be noted that other sea ice prediction approaches have outperformed persistence (e.g., Tivy et al.,2007; Shröder et al., 2014; Yuan et al., 2016; Petty et al., 2017; and Bushuk et al., 2018).  Some of these studies have used statistical methods informed by other predictors (e.g., Lindsay et al., 2008; Tivy et al., 2011),  some have used the perfect model approach (e.g., Blanchard-Wrigglesworth et al., 2011), and some have made use of initialized hindcasts (e.g., Bushuk et al., 2018). Nevertheless, persistence derived skill provides a baseline for the measurement of forecast skill achieved by these other approaches and, based on the results in Section 5, persistence of departures from the trend lime can be a challenging competitor at forecast ranges of months to seasons.

While the variance explained by simple anomaly persistence at lead times of several seasons and also by persistence of detrended anomalies at lead times of a month or two is statistically significant, statistical significance does not equate to usefulness. Potential users of sea ice forecasts include local communities engaging in offshore subsistence and travel activities, marine transport companies, offshore resource extraction, and the tourism industry.  The relatively small fractions of variance predictable several months in advance using detrended data (Figures 6-9) will likely leave uncertainties that are too great for many users.  However the trend-derived skill, which can represent 50% or more of the variance, may enable decisions if the interannual variations superimposed on the trend represent acceptable risks for users of sea ice forecasts.

**Acknowledgments**

Funding for this work was provided by the Climate Program Office of the National Oceanic and Atmospheric Administration through Grants NA16OAR4310162 and by the National Science Foundation through Grant OPP-1749081.  Florence Fetterer was supported by the CIRES/NOAA Cooperative Agreement, NOAA Grant NA15OAR4320137. We thank the two reviewers for their many comments and suggestions, which have led to a more rigorous presentation.  We also thank Larry Hamilton of the University of New Hampshire for initiating the comparison of the skill of the persistence-based forecasts and the Sea Ice Outlook.

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

use the HSIA because it extends the record 26 years back in time before the start of the satellite
passive microwave record.  While the sources of the ice concentration data in the HSIA are the
same as in G10010, a notable advantage of the HSIA is its weekly temporal resolution (vs. the
monthly resolution of G10010).  The HSIA also has a spatial resolution of ¼° latitude by ¼° degree
longitude.  Because of the weekly time resolution, the distance metrics (3)-(5) of the BSI are
truncated to the nearest week.  Similarly, the distance metrics (1) and (2) are truncated to the
nearest 27.8 km (15 n mi).  One of the within-month dates of the HSIA grids is the 15$^{th}$ of each
month, so no temporal interpolation is necessary for metrics (1) and (2).  The reconstructed values
of the BSI are listed in Table A1.


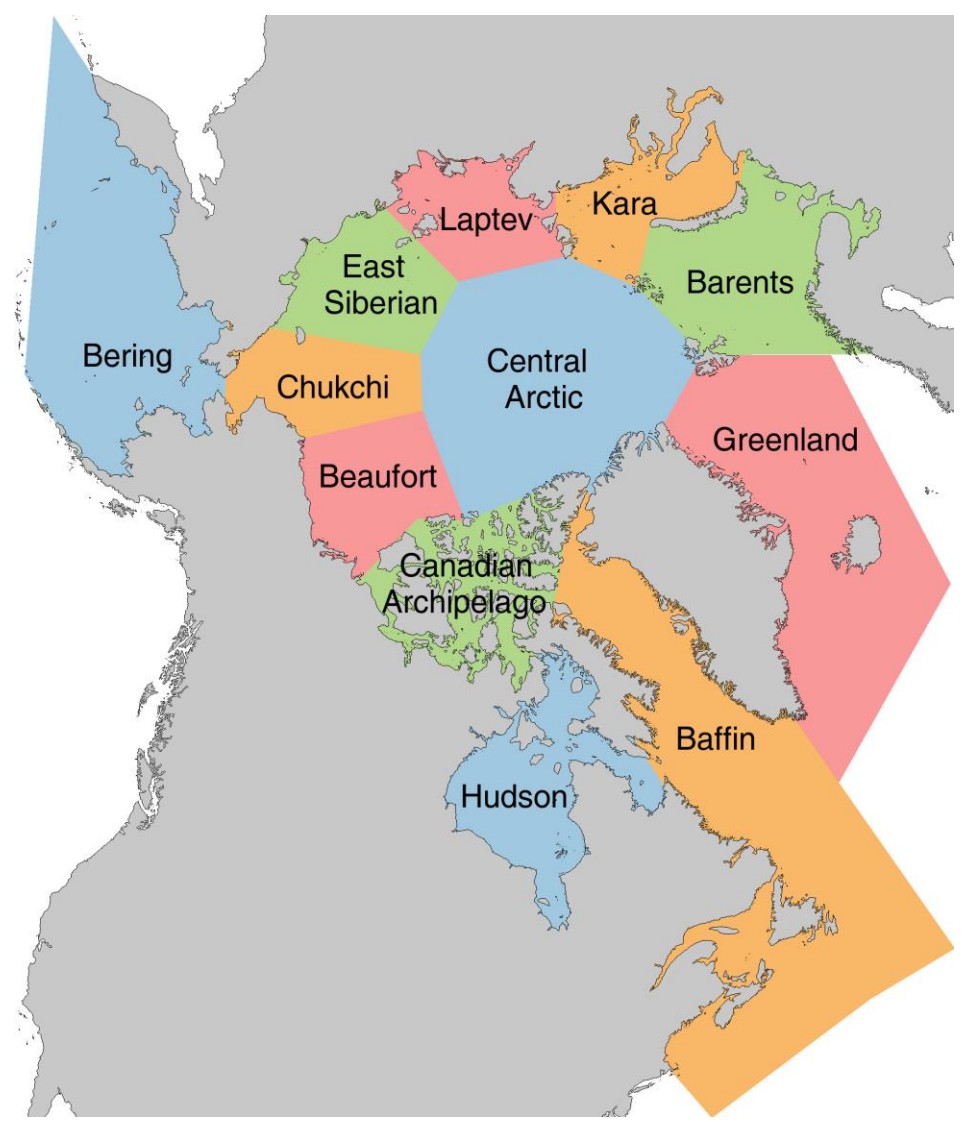

**Figure 1.** The MASIE subregions used in the study (NIC and NSIDC, 2010).

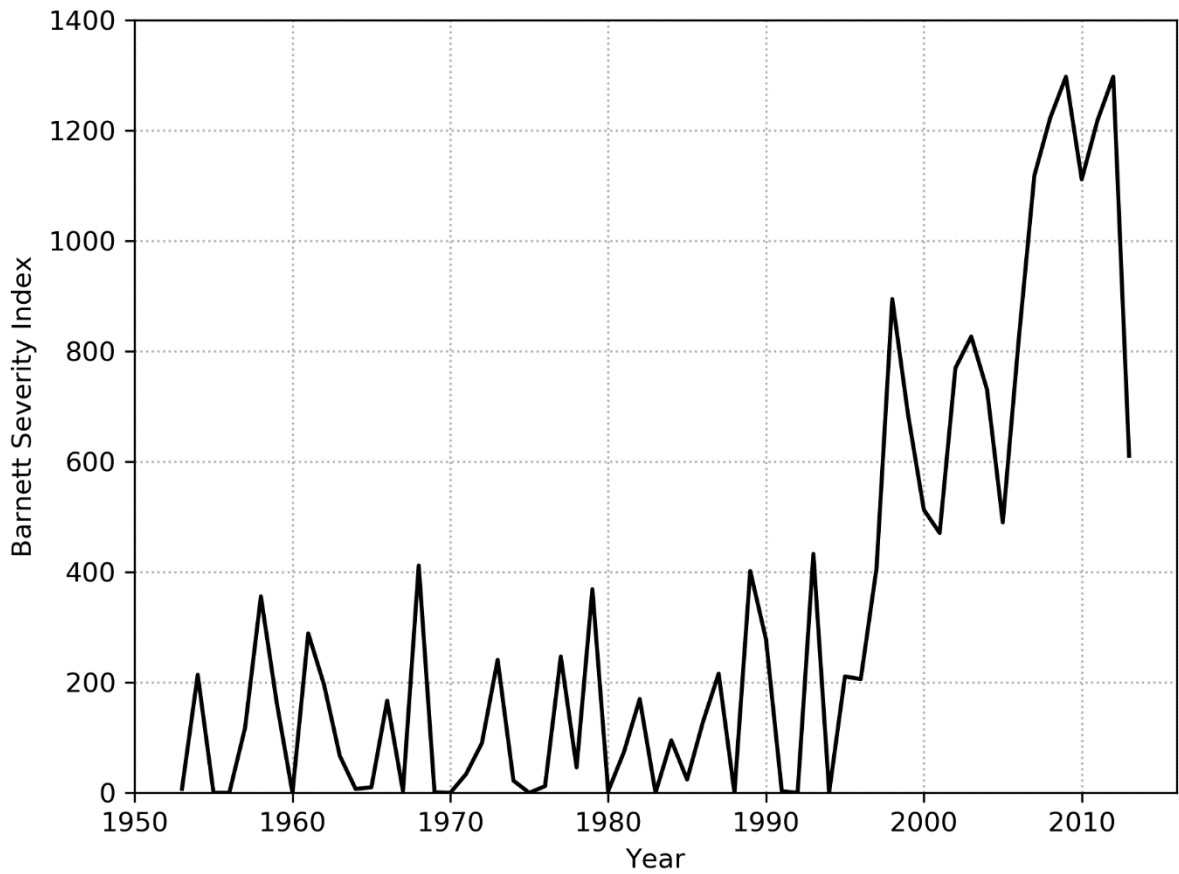

**Figure 2.**  Time series of the Barnett Severity Index (BSI), 1953-2013.

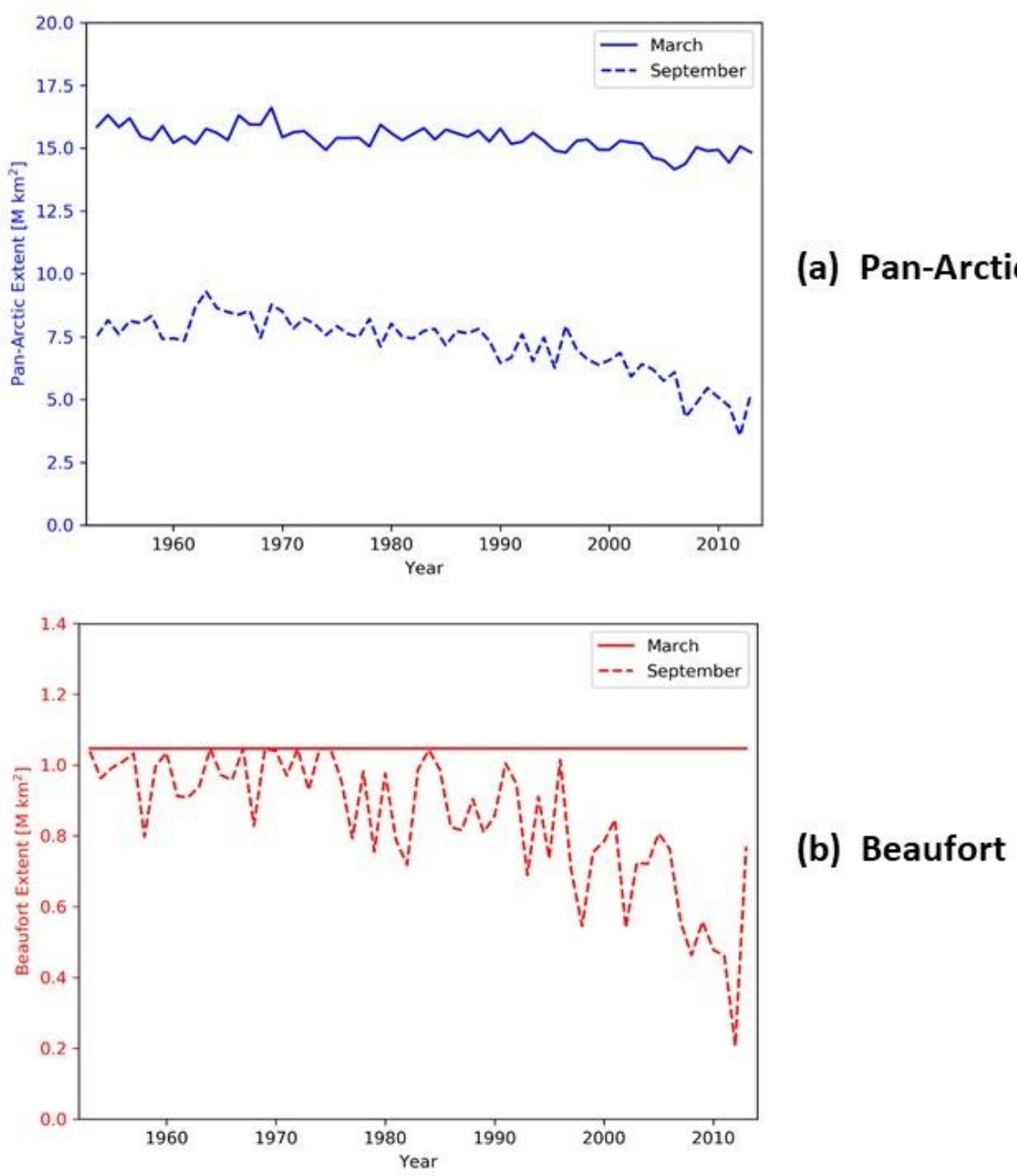


**Figure 3.** **(a)** Total Arctic sea ice extent and (b) the extent of ice in the Beaufort Sea during

March( solid lines) and September (dashed lines) .


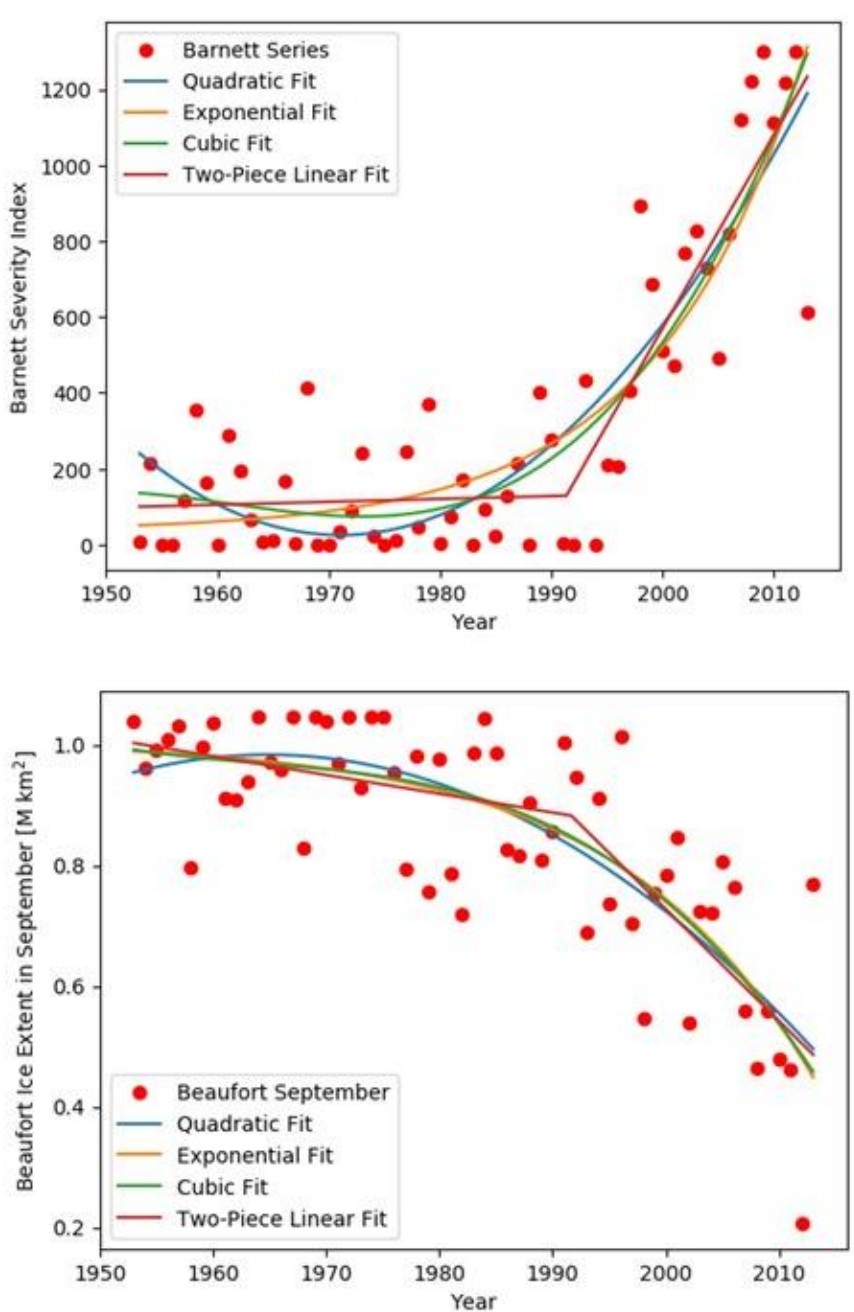


**Figure 4.** Examples of different fit methods (see legend) applied to the BSI (upper panel) and
the September Beaufort ice extent time series (lower panel).




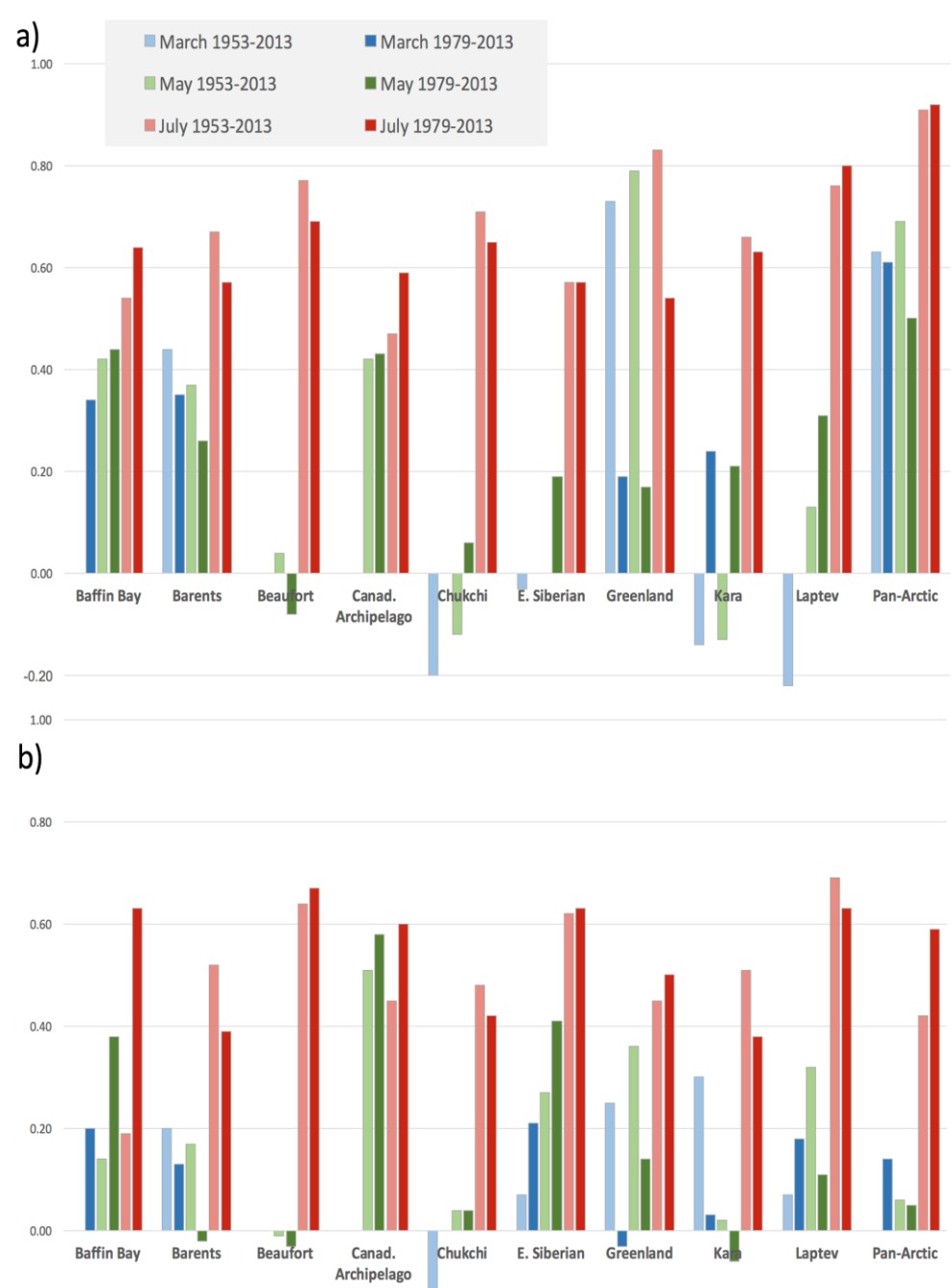

**Figure 5.** Correlations of September ice extents in individual seas with ice extent in the same region in March (green bars), May (blue bars) and July (red bars) bases on (a) non-detrended data and (b) detrended data. Correlations are also shown for Pan-Arctic extent (far right in each panel). The correlations are based on non-detrended data. For each color, light-colored bars are for 1953-2013 and dark-colored bars are for 1979-2013. The absence of a bar indicates a correlation of zero. The 95% significant levels for the longer and shorter samples are 0.26 and 0.33, respectively.

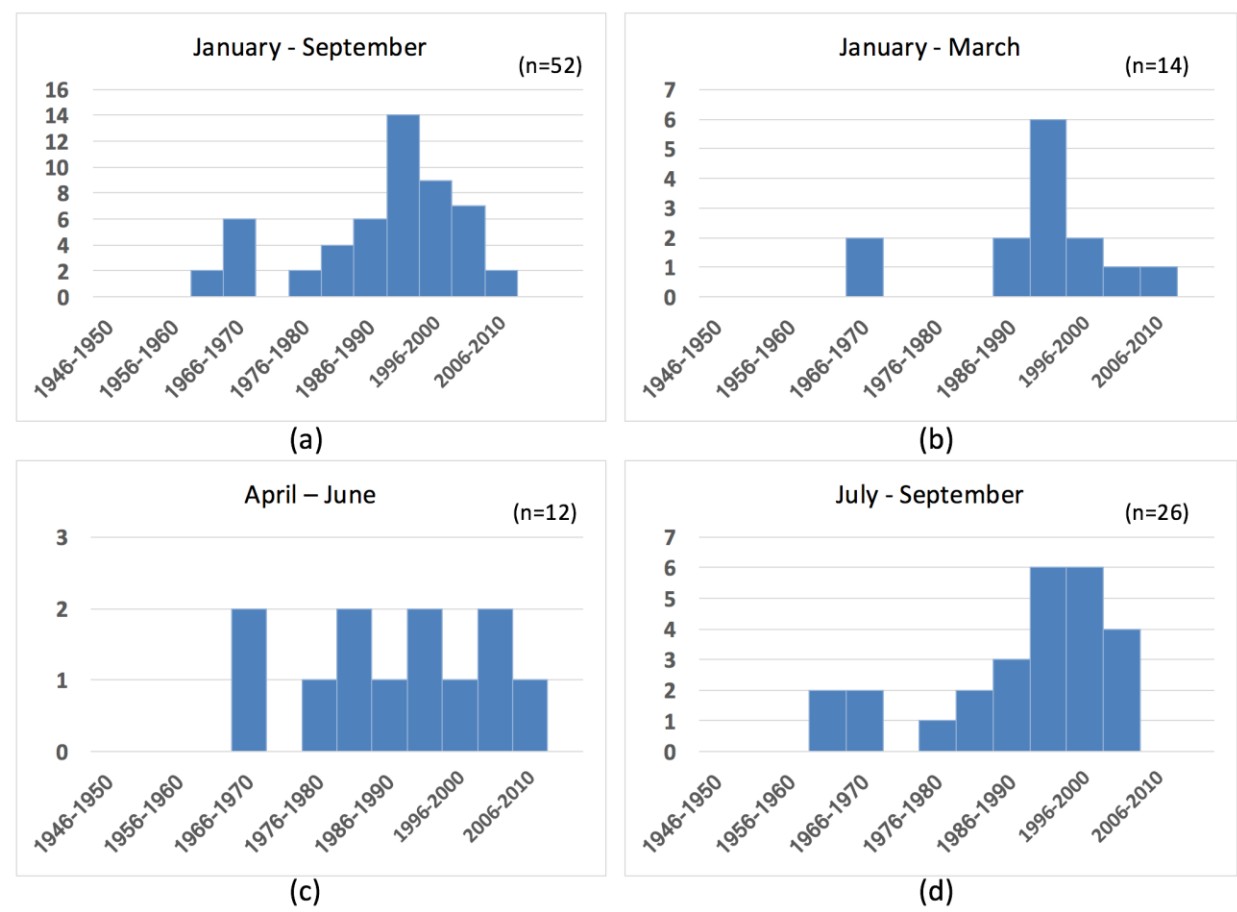

**Figure 6.** The distribution of break-point years across all regions for (a) January-September and its three subperiods: (b) January-March, (c) April-June, (d) July-September). Only cases for which detrending using two lines, rather than one, reduced the root-mean-squares error by 5% or more are included. Note that y-axes have different scales.

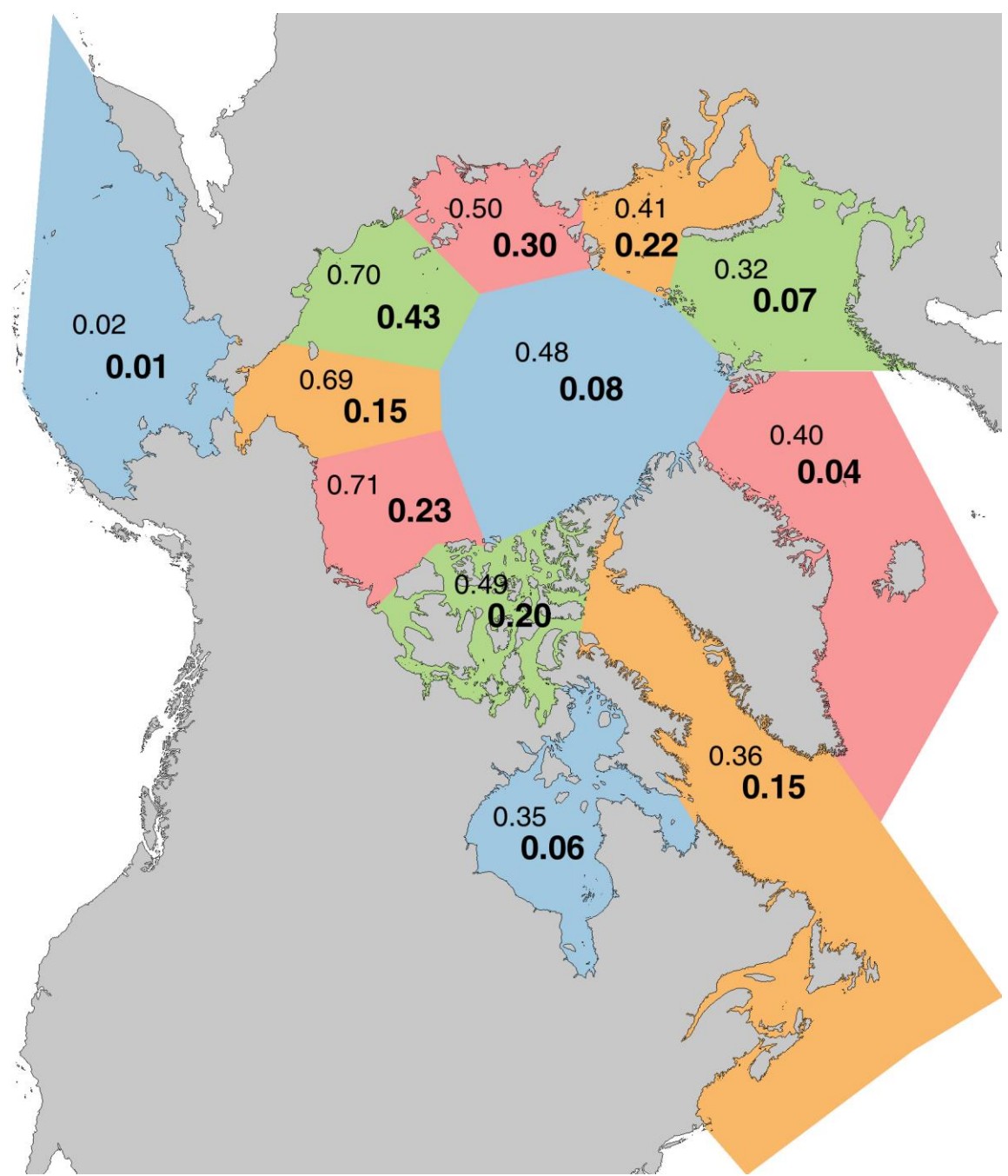

**Figure 7.** Squares of correlations ($R^2$) between September pan-Arctic ice extent and September

regional ice extent based on ice extents including trends (upper numbers in normal

font) and detrended (lower numbers, bold font). The 95% significance thresholds for

the correlations range from 0.26 with no autocorrelation (generally the case for

detrended data) to 0.38 with an autocorrelation of 0.4; the corresponding $R^2$ thresholds

are 0.07 and 0.14.

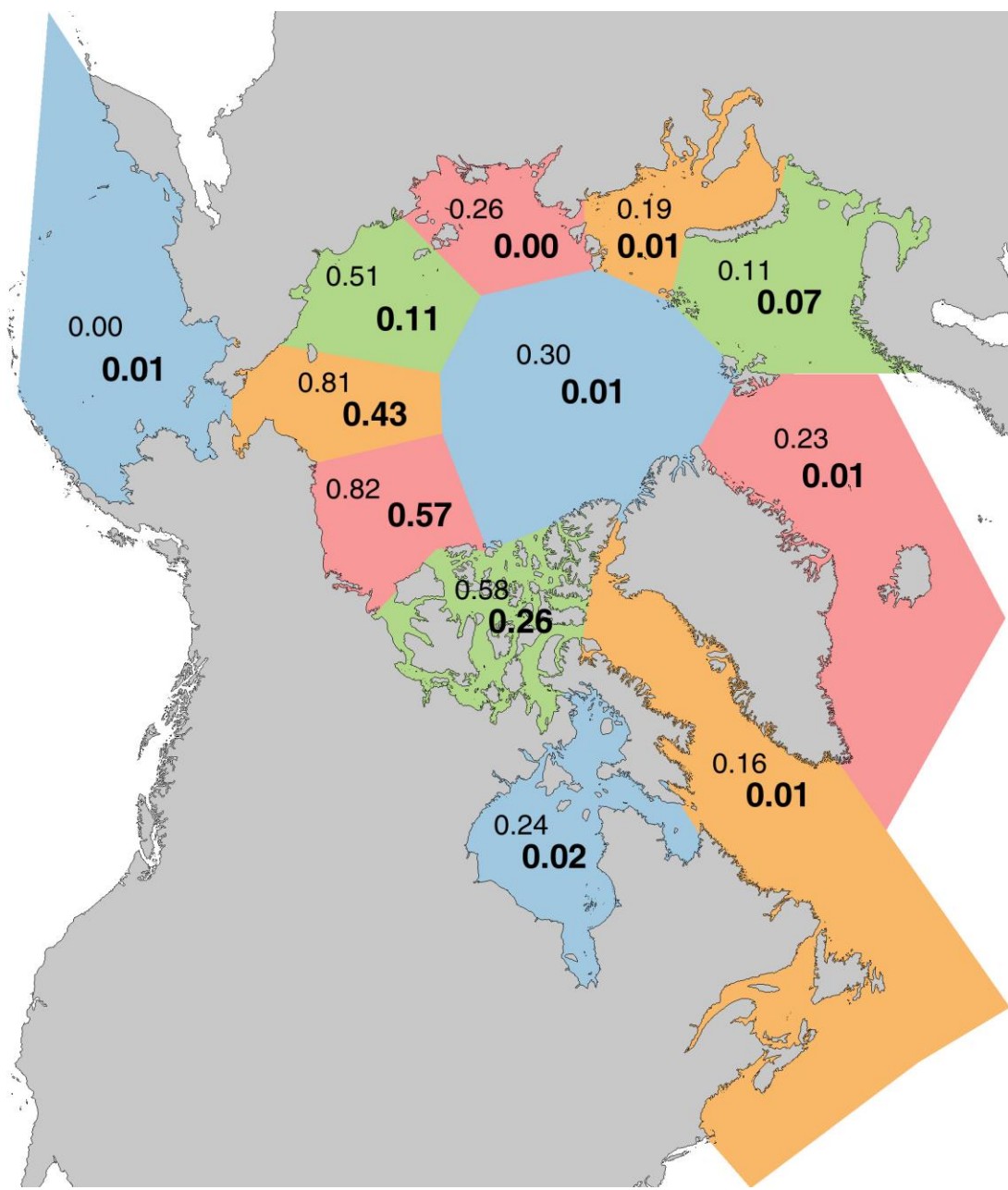

733

734

**Figure 8.** As in Figure 7, but for squares of correlations between the annual BSI and September regional ice extents based on raw (not detrended) time series (upper numbers) and detrended time series (lower numbers, bold font). The 95% significance thresholds for the $R^2$ values range from 0.07 with no autocorrelation (generally the case for detrended data) to 0.14 with an autocorrelation of 0.4.

740

741

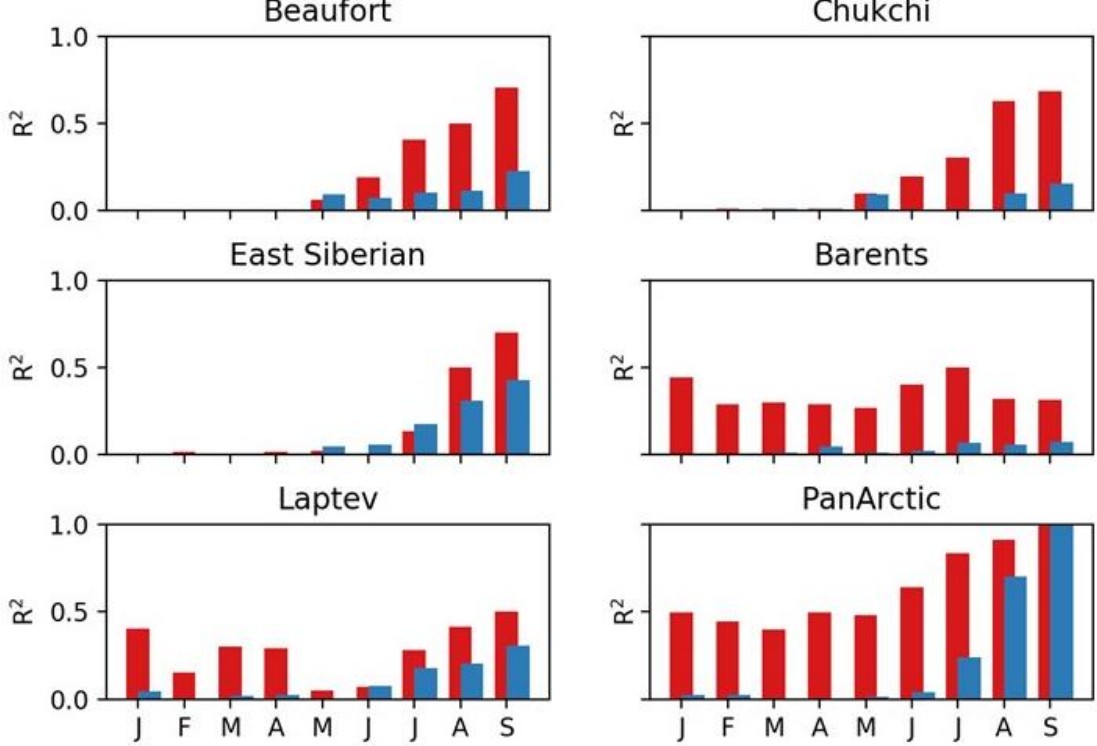

742

**Figure 9.** Examples of variances of September pan-Arctic ice extent explained by correlations with antecedent regional ice extent in individual calendar months from September back to January (pan-Arctic extent lagging by 0, 1, 2, …, 8 months). Explained variances are plotted as fractions of explained variance (squares of correlations). Red bars are values with trends included, blue bars are correlations after removal of trends. The 95% significance thresholds for the $R^2$ values range from 0.07 with no autocorrelation (generally the case for detrended data) to 0.14 with an autocorrelation of 0.4.

750

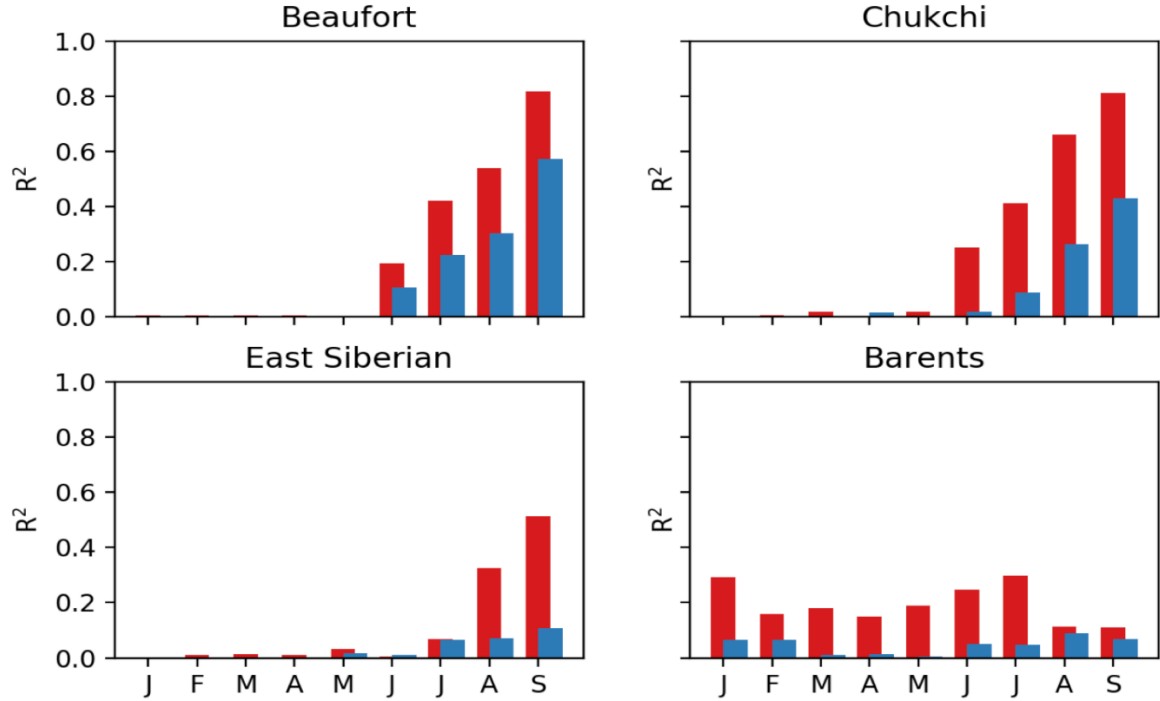

**Figure 10.** Examples of variances explained by correlations between the Barnett Severity Index and regional ice extent in individual calendar months from September back to January (BSI lagging by 0, 1, 2, …, 8 months). Explained variances are plotted as fractions of explained variance (squares of correlations). Red bars are values with trends included, blue bars are correlations after removal of trends. Significance thresholds as in Figure 9.

| Region | Jan | Feb | Mar | Apr | May | Jun | Jul | Aug | Sep |
|---|---|---|---|---|---|---|---|---|---|
| **Baffin-St. Lawrence** | 0.08 | 0.06 | 0.02 | 0.16 | 0.32 | 0.49 | 0.61 | 0.52 | 0.36 |
| **Barents** | 0.45 | 0.29 | 0.30 | 0.29 | 0.27 | 0.41 | 0.50 | 0.32 | 0.32 |
| **Beaufort** | 0.41 | 0.41 | 0.41 | 0.41 | 0.06 | 0.19 | 0.41 | 0.50 | 0.71 |
| **Bering** | 0.00 | 0.00 | 0.01 | 0.00 | 0.01 | 0.12 | 0.08 | 0.01 | 0.02 |
| **Canadian Archipelago** | 0.41 | 0.41 | 0.41 | 0.41 | 0.00 | 0.13 | 0.09 | 0.52 | 0.49 |
| **Central Arctic** | 0.21 | 0.11 | 0.18 | 0.01 | 0.02 | 0.02 | 0.15 | 0.07 | 0.48 |
| **Chukchi** | 0.00 | 0.01 | 0.01 | 0.01 | 0.10 | 0.20 | 0.31 | 0.63 | 0.69 |
| **East Siberian** | 0.00 | 0.02 | 0.01 | 0.02 | 0.02 | 0.00 | 0.14 | 0.50 | 0.70 |
| **Greenland** | 0.47 | 0.53 | 0.50 | 0.48 | 0.43 | 0.41 | 0.45 | 0.29 | 0.40 |
| **Hudson** | 0.05 | 0.41 | 0.41 | 0.26 | 0.03 | 0.32 | 0.66 | 0.37 | 0.35 |
| **Kara** | 0.00 | 0.03 | 0.11 | 0.04 | 0.10 | 0.09 | 0.44 | 0.42 | 0.41 |
| **Laptev** | 0.40 | 0.15 | 0.30 | 0.29 | 0.05 | 0.07 | 0.28 | 0.42 | 0.50 |
| **Pan-Arctic** | 0.50 | 0.44 | 0.40 | 0.50 | 0.48 | 0.64 | 0.84 | 0.91 | 1.00 |

**Table 1**. Correlations between monthly regional ice extent and pan-Arctic ice extent expressed as explained variance ($R^2$). Cases where at least 10% of the variance in pan-Arctic ice extent is explained by regional ice extent in a given antecedent month are highlighted with bolded region names. Levels of shading of boxes denote values exceeding 0.10, 0.20, 0.30,…

| Region | Jan | Feb | Mar | Apr | May | Jun | Jul | Aug | Sep |
|---|---|---|---|---|---|---|---|---|---|
| **Baffin-St. Lawrence** | 0.09 | 0.04 | 0.08 | 0.06 | 0.01 | 0.00 | 0.03 | 0.16 | 0.15 |
| Barents | 0.00 | 0.01 | 0.01 | 0.05 | 0.01 | 0.02 | 0.07 | 0.06 | 0.07 |
| **Beaufort** | 0.05 | 0.05 | 0.05 | 0.05 | 0.10 | 0.08 | 0.11 | 0.11 | 0.23 |
| Bering | 0.01 | 0.01 | 0.08 | 0.03 | 0.02 | 0.00 | 0.01 | 0.01 | 0.01 |
| **Canadian Archipelago** | 0.05 | 0.05 | 0.05 | 0.05 | 0.01 | 0.02 | 0.02 | 0.16 | 0.20 |
| **Central Arctic** | 0.02 | 0.02 | 0.11 | 0.03 | 0.02 | 0.04 | 0.07 | 0.00 | 0.08 |
| **Chukchi** | 0.00 | 0.00 | 0.01 | 0.01 | 0.10 | 0.00 | 0.00 | 0.10 | 0.15 |
| **East Siberian** | 0.00 | 0.00 | 0.00 | 0.00 | 0.05 | 0.06 | 0.18 | 0.31 | 0.43 |
| Greenland | 0.06 | 0.04 | 0.09 | 0.07 | 0.03 | 0.06 | 0.04 | 0.00 | 0.04 |
| **Hudson** | 0.00 | 0.05 | 0.05 | 0.01 | 0.05 | 0.01 | 0.11 | 0.07 | 0.06 |
| **Kara** | 0.01 | 0.03 | 0.03 | 0.04 | 0.00 | 0.18 | 0.12 | 0.13 | 0.22 |
| **Laptev** | 0.05 | 0.00 | 0.02 | 0.02 | 0.01 | 0.08 | 0.18 | 0.21 | 0.30 |
| **Pan-Arctic** | 0.03 | 0.02 | 0.00 | 0.00 | 0.01 | 0.04 | 0.24 | 0.70 | 1.00 |

**Table 2.** Correlations between detrended monthly regional ice extent and detrended September pan-Arctic ice extent expressed as explained variance ($R^2$). Cases where at least 10% of the variance in September pan-Arctic ice extent is predictable by regional ice extent in a given antecedent month are highlighted with bolded region names. Shading of boxes is as in Table 1.

| Year | BSI | Year | BSI |
|------|-----|------|-----|
| 1953 | 7 | 1984 | 95 |
| 1954 | 213 | 1985 | 24 |
| 1955 | 0 | 1986 | 178 |
| 1956 | 0 | 1987 | 216 |
| 1957 | 117 | 1988 | 0 |
| 1958 | 356 | 1989 | 402 |
| 1959 | 163 | 1990 | 278 |
| 1960 | 0 | 1991 | 3 |
| 1961 | 289 | 1992 | 0 |
| 1962 | 195 | 1993 | 434 |
| 1963 | 66 | 1994 | 1 |
| 1964 | 7 | 1995 | 211 |
| 1965 | 10 | 1996 | 206 |
| 1966 | 167 | 1997 | 407 |
| 1967 | 3 | 1998 | 895 |
| 1968 | 412 | 1999 | 685 |
| 1969 | 1 | 2000 | 513 |
| 1970 | 0 | 2001 | 471 |
| 1971 | 34 | 2002 | 770 |
| 1972 | 90 | 2003 | 827 |
| 1973 | 240 | 2004 | 731 |
| 1974 | 22 | 2005 | 490 |
| 1975 | 0 | 2006 | 819 |
| 1976 | 13 | 2007 | 1119 |
| 1977 | 247 | 2008 | 12239 |
| 1978 | 46 | 2009 | 12989 |
| 1979 | 368 | 2010 | 1112 |
| 1980 | 3 | 2011 | 1219 |
| 1981 | 74 | 2012 | 1298 |
| 1982 | 170 | 2013 | 611 |
| 1983 | 0 | | |

**Table A1**. Yearly values of the Barnett Severity Index (BSI). Source: Rebecca Rolph, Geophysical Institute, University of Alaska, Fairbanks.

