# Peer review of "Seasonal sea ice prediction based on regional indices"

_The Cryosphere, 2018_

## Referee Comment (RC1) · Anonymous Referee #1 · 4 Oct 2018

**Overview and broad comments**

This study by Walsh et al uses a long historical record of sea ice coverage data to estimate the amount of variance explained in two (mainly September) quantities, the Beaufort Sea Index and pan-Arctic sea ice extent, by the sea ice extent in different Arctic sub-regions. They consider both concurrent correlations and lagged correlations of these quantities with sea ice conditions for previous months, and separate these correlations into a total and interannual estimation. They argue that a piece-wise linear trend is most appropriate for detrending the different time series for estimating the latter.

[Figure]

The noteworthy pieces of information to come out of this study are (1) a piece-wise linear trend is the best option for detrending the time series considered in the study, (2) the break-point year emergent in that analysis which can be interpreted as an acceleration of the ongoing negative trend in sea ice cover is in the mid- to late- 1990's, (3) that interannual variability in the BSI can be explained by June sea ice coverage in the Beaufort Sea and by July coverage in the Chukchi Sea, (4) consistent with other studies the September pan-Arctic ice extent has significant autocorrelation back to July (about two months), and (5) that the Laptev and East Siberian Seas explain the most concurrent correlation in September pan-Arctic SIE.

As it stands though, in my opinion the paper requires substantial revisions and additional analyses before being re-submitted.

The key goal of the study seems to be to provide baseline metrics against which sea ice prediction studies can be evaluated. However, that baseline has already been established for one of the two predictands considered in this study, pan-Arctic SIE, in several other studies based on autocorrelation (i.e. persistence). No reference to these other studies is ever made. The one thing that separates the result about the pan-Arctic SIE predictand in this study from others is that a very long historical record was used here, but that point should be emphasized when motivating the study (it's currently not mentioned). The rest of the lagged correlation analysis for pan-Arctic SIE, which was between it and the SIE for the various subregions, didn't yield higher correlations than lagged pan-Arctic sea ice extent itself. So, for evaluating prediction skill, is it not the autocorrelation of pan-Arctic SIE that is the important baseline to beat? What therefore is the motivation to consider the lagged correlation analysis with the different sub-regions?

With respect to the BSI predictand, that analysis seems okay and is fine to report on, but I'm not sure all of the information presented (particularly in the tables; see specific comments below) is needed to reach the conclusions made.

I was also a bit surprised (and disappointed) to see that only these two predictands were considered if the goal is to provide baseline skill numbers, especially considering the first predictand has already received considerable attention in other studies. Like in the supplementary material of Bushuk et al 2018 (reference below), it would be good to extend the analysis to treat all of the different regions as separate predictands and compute the autocorrelations for each of those regions. This would help put the results of the Bushuk et al study into context, as they used a shorter and different observational dataset than is used here. As of now, and considering some of my suggestions to remove certain parts of the results section (see specific comments below), this additional analysis would help strengthen that section and would provide useful baseline numbers for future studies.

I was missing from the introduction a sufficient argument for the need for this study in the context of current literature on sea ice prediction. The additional references to, and discussion of some key studies listed below are needed to place the goals of this study and its findings into perspective. Specifically here are studies on persistence/autocorrelation of observed and modeled pan-Arctic and regional sea ice extent:

Blanchard-Wrigglesworth, Edward, et al. "Persistence and inherent predictability of Arctic sea ice in a GCM ensemble and observations." Journal of Climate 24.1 (2011): 231-250.
– Includes estimates of autocorrelation for pan-Arctic SIE and SIA in both observations and models. Also includes contributions of lagged Beaufort Sea sea ice thickness to september SIA and SIE variability.

– This paper was referenced in the study, but it's odd that only the SST re-emergence result was mentioned.

Sigmond, M., et al. "Seasonal forecast skill of Arctic sea ice area in a dynamical forecast system." Geophysical Research Letters 40.3 (2013): 529-534.
– Their Fig. 3 shows persistence forecasts of SIE from two observational datasets.

Bushuk, Mitchell, et al. "Regional Arctic sea–ice prediction: potential versus operational seasonal forecast skill." Climate Dynamics (2018): 1-23.
– In their supplementary material, they show autocorrelations for individual regions from observations and compare against models.

Throughout the paper, the word "predictability" is used somewhat misleadingly. It should be made clear early onthat what is being referred to as predictability throughout the rest of the paper is a very specific estimate of predictability – that is, the predictability of a certain sea ice coverage quantity based on the lagged cross-correlation or auto-correlation (often referred to as memory or persistence) between that quantity and other sea ice coverage quantities. Otherwise, it sounds like the authors are referring to predictability in general, which encompasses all sources of predictability and estimates of theoretical predictability limits.

Guemas, Virginie, et al. "A review on Arctic sea–ice predictability and prediction on seasonal to decadal time–scales." Quarterly Journal of the Royal Meteorological Society 142.695 (2016): 546-561.
– Provides a review on sources of predictability, including memory/persistence of sea ice coverage and thickness. See references therein.

Bushuk, Mitchell, et al. "Regional Arctic sea–ice prediction: potential versus operational seasonal forecast skill." Climate Dynamics (2018): 1-23.

– Focuses on regional predictability.

The de-trending analysis could be expanded on. The 1979-onward period is what is commonly used in prediction studies, and there is an ongoing debate in the literature about what de-trending method is most appropriate for pan-Arctic SIE, particularly when the most recent few years are included. Extending the de-trending analysis to focus also on this shorter period would be helpful for future prediction studies. While most authors choose to fit a linear trend over that period, Dirkson et al 2017 suggested a quadratic fit; others suggest de-trending with a high-pass filter but this has the unfortunate effect of removing the first or last sample from the data over an already short period. It would be worthwhile, and would add to the paper, if the authors could make a case for the piece-wise linear trend over this shorter period if indeed it is much better than a linear or quadratic fit. Also, as prediction studies are beginning to focus more on regional sea ice prediction than on pan-Arctic SIE/SIA, it would be helpful to know what choice is most appropriate for the different regions and whether this choice depends on the month or season. For instance, Bushuk et al 2017 and Dirkson et al 2017 argued that linear de-trending was sufficient for regional SIE and local SIC over a similar period considered here.

Dirkson, Arlan, William J. Merryfield, and Adam Monahan. "Impacts of sea ice thickness initialization on seasonal Arctic sea ice predictions." Journal of Climate 30.3 (2017): 1001-1017.

Bushuk, Mitchell, et al. "Skillful regional prediction of Arctic sea ice on seasonal timescales." Geophysical Research Letters 44.10 (2017): 4953-4964.

**Specific Comments**

- It should be stated in the abstract early on (before L11-13) the prediction method that will be used in this paper.

- L19: "statistical predictability" should be replaced with "statistical skill..." based on said approach. It's not statistical predictability in the sense of a theoretical limit, which isn't directly possible to estimate .

- L17-18; Which month(s) are being referred to here?

- L31-32; Statement requires a reference.

- L37-39; Should cite:
  Maslanik, James, et al. "Distribution and trends in Arctic sea ice age through spring 2011." Geophysical Research Letters 38.13 (2011).

- L47-50; This positive trend in the Bering Sea in winter is disappearing if more recent years are taken into account... Ah, this is stated in the conclusions, but is probably more appropriate to place here instead.

- The departures are also affected by antecedent sea ice conditions themselves. Please see the references given in the general comments above related to persistence.

- L79-82; This is not correct. "Ice-ocean model" implies that both the ice and ocean evolve freely as determined by the model (only the atmospheric forcing is required). Also, the statement ignores the fact that fully coupled models, which determine both the atmospheric and ocean/ice conditions prognostically, are now used more often for sea ice prediction. These models are also limited by the chaotic nature of the climate system, but this is typically accounted for by running ensembles. There is important literature on predictability using said models that isn't referenced or discussed here. Please see general comment above highlighting predictability literature.

- L83-84; It's difficult to see how this paper is an "extension" of Drobot 2003; while this study also focuses on the Beaufort and Chukchi Seas, and extends an analysis of statistical prediction skill in that region based on more recent observational data, the analysis done in Drobot 2003 (statistical prediction using multiple linear regression with many more predictors) is not repeated here.

- L88-92; The caution raised in the Drobot 2003 study I think definitely deserves attention given Arctic sea ice and Arctic climate have changed, as stated. Although this study distinguishes between prediction of sea ice coverage with and without the trend, as written it sounds like the authors will carry out an analysis similar to Drobot 2003, and determine the impact of the changing conditions on the statistical relationships found in that study, which is not what is done here.

- L95-96; This is incorrect. While the Drobot 2003 study didn't consider the effects of detrending as stated, Blanchard-Wrigglesworth et al 2011 did. Specifically they detrended the model projections of SIE by subtracting the ensemble mean at each point in time (removing the forced signal). Additionally, they detrended the SIE observations by subtracting the long-term linear trend.

- L97-100; I find this overview too vague and is broader in scope than what is actually done in the paper. I think it would be more accurate and helpful to the reader if the authors provided, in order of appearance in the paper, what will specifically be done.

- L102; "seasonal climatologies, persistence, and trend" of what? Should clarify that you are referring to these quantities for sea ice coverage itself.

- L104; "ice-ocean models" should be replaced with "dynamical models". Ice-ocean models are a specific subset of these, but fully-coupled models are used more often.

- L105; The Sea Ice Outlook was managed by the Sea Ice Prediction Network, and starting in 2018 became managed by the Sea Ice Prediction Network–Phase 2. The Sea Ice Outlook is not a previous name for Sea Ice Prediction Network. See https://www.arcus.org/sipn/sea-ice-outlook.

- L118; Suggest changing "maps" to "gridded records". Presumably what is being referred to here is data in digitized form and not the presentation of the data literally as maps.

- L124-127; Say the product name here. The product is described in the next sentence without actually saying explicitly that it is used.

- L154-155; While this is true, previous studies have shown that there are shorter persistence time scales for pan-Arctic SIE than for pan-Arctic SIA due to high frequency dynamical influences that change SIE, but not SIA (Blanchard-Wrigglesworth et al 2011; their Fig. 14). This should be mentioned here.

- L157ff; Was there any interpolation done to get the G10010 dataset onto the same grid used to define the MASIE sub-regions?

- L182; Somewhere early on in the methods section it should be stated what years are considered in the study.

- L191-192; Please clarify. I think what is meant is that the presence of a trend in a time series inflates forecast skill when using the anomaly correlation coefficient to assess that skill. It should be noted though that for distance metrics like the root mean square error, skill can actually be inflated by detrending the data if the two time series have different magnitudes of trends.

- L198-202; please refer to general comment on de-trending, but this paragraph would be a good place to reference choices made in other sea ice prediction studies.

[Figure]

- L209ff; As of now, I find the description of the fitting methodology a bit hard to follow. It could be more straight-forward both in terms of its description and implementation. If one thinks of the function as a piece-wise linear trend defined by

$$y = a_1 x + b_1, \quad x < x_b \tag{1}$$

and

$$y = a_2 x + b_2, \quad x > x_b \tag{2}$$

where $x_b$ is the breakpoint year, continuity of $y$ at $x_b$ can be ensured by letting $b_2 = (a_1 - a_2)x_b + b_1$. Substituting $b_2$ into the equation yields 4 parameters to be estimated $(a_1, b_1, a_2, x_b)$, which can be done using the 'curve_fit' function. This avoids having to also use both the 'curve_fit' and 'lin_regress' function, as the parameters found by 'curve_fit' can be used to describe the two lines completely. Regardless of which method is used, the equation(s) used in 'curve_fit' should be shown in the paper.

- L227-229; This was just stated on L207-208.

- L239ff; This is an interesting result, but is it necessary to include the break-point year for all calendar months from January-September in Figure 5? The piece-wise equation seems like a sensible approach to determine a breakpoint year in the summer months (except maybe in the central Arctic). However, for winter and spring months is it really the case that a two-piece linear trend yields a better mean squared error than a simple linear trend? If not, then the breakpoint year for those months won't carry any significant meaning other than that it's an emergent parameter you get from fitting the two-piece linear trend. Is that perhaps what is being seen in Fig. 5 for years before the 1990's and after the early 2000's? Breaking down the information in Fig. 5 for each month and region would make the results easier to interpret and more informative.

- L250; Is the reason for showing Figure 6 not also to see what regions contribute the most explained variance to pan-Arctic SIE?

- L256-260; Why say the significance level for this specific autocorrelation value? Is it the maximum autocorrelation value for all of the samples considered?

- L260-262; This is the significance value when the sample has an autocorrelation of zero. Do the detrended time series have no autocorrelation? Also, this statement is true for most regions, but not all according to Fig. 6.

- L263-265; This is true when the trend is included, but not when the time series are detrended according to Fig 6. Please clarify which is being referred to here. Can the authors speculate at all why it is the East Siberian Sea and Laptev Sea explain the most synchronous interannual variance in pan-Arctic SIE? Is it the region that has the most interannual variability in September?

- L266-275; Is Figure 7 really necessary? It seems reasonable to consider the contributions of variability in different regions to variability in pan-Arctic SIE, but is there any physical basis for thinking that regions far away from the Beaufort Sea would show any explained variance in the BSI, apart from the trend? It's worthwhile to know how far from the Beaufort Sea variations matter, but those regions are shown in Fig. 9 already for lag 0.

- Are tables 1-4 necessary?

For Tables 1 and 3, when the full time series are compared (non-detrended), the fact that there will be correlation between the different regions and the 2 predictands when both predictor and predictand contain trends is not really a surprise, is it? For Tables 2 and 4, the only relevant information to prediction of the BSI is limited to a few (expectantly nearby) regions, and for the most part to a short number of lag months. For those areas and lags where there is any

explained variance over 10% , that information is already plotted in Figs. 8 and 9. Why not just say when describing Figs 8 and 9 that no explained variance values greater than 10% were found in other regions?

If an argument to keep the tables is made, the shading scheme should be explained.

- L284ff and Figure 8; I'm surprised to see the values of 0.41 and 0.05 for the Beaufort Sea in Fig. 8 (and identical values for the Canadian Archipelago in the tables 1, 2) for lag months January-April. How can this be if sea ice extent is not variable in the Beaufort Sea and the Canadian Archipelago in the winter?

- Figure 8; I think it makes sense to also show the autocorrelation for September pan-Arctic SIE (recognizing it is in Tables 1 and 2; see comment above). It would put the contributions of these other regions into context for prediction purposes. According to Table 2, Pan-Arctic SIE contributes at different lags contributes more to September SIE than any individual region. Should that result itself be mentioned as the most significant (albeit arguably expected) result in terms of prediction? This result should be compared against the studies mentioned in the general comments, but it seems to agree with them. Also, the lagged Laptev Sea SIE contributes more to September Pan-Arctic SIE than the Beaufort Sea, so why not show it too?

  A physical explanation for Laptev Sea can be found in Williams et al 2016:

  Williams, James, et al. "Dynamic preconditioning of the minimum September sea-ice extent." Journal of Climate 29.16 (2016): 5879-5891.

- L288-291; "...reaching zero by...". Near zero, not zero exactly. It's also odd that these are not zero exactly when ice covers these regions completely through April or May.

- L319-320; Doesn't the BSI contain information from earlier months than September? That is, it's not a strictly September prediction metric.

- L336-338; Specify that this is a general finding; it's certainly not true for all "ice extents" considered.

- L338-340; "(and even years, given the multidecadal scales of the trends)"... While probably true, this isn't a direct conclusion/finding of this study.

- 342-345; The study being referenced here is based on six years of hindcasts which were created by averaging over many different techniques/models, and the results are not reflective of the predictive skill shown by several other prediction studies. To put the results of this paper into context, it should be stated that numerous other studies have shown higher skill than anomaly persistence forecasts (which is essentially the method used here). See references in the introduction of Bushuk et al 2018 (reference in general comments).

- L355; The influence of trend predictive skill is well recognized in the sea ice prediction community, and skill results based on detrending are commonly presented in sea ice prediction studies, so I'm not sure I see how challenge (1) follows from this particular study.

**Technical corrections**

- Figure 3: I think it would be easier to see and compare magnitudes of the anomalies and trends if Fig 3 were split into two sub-panels: one for the Beaufort and one for pan-Arctic SIE. The sub-panel for pan-Arctic SIE could have double y-axis (one for March, one for September) so that there isn't such a large vertical space between lines.

- L194-197; Sentence is worded awkwardly, specifically the "Because..., so ...".

- Figure 4: It looks like there is a kink in the piece-wise linear functions near the breakpoint. This shouln't be there.

- L255; "accrual".. what is accruing here? Why not just say "corresponding to correlations..."

- L257 and 258; typo? "a 60-year samples" should be "a 60-year sample"

- L296; Change to ..."when sea ice extent" for the Beaufort and East Siberian Seas are predictors. The seas themselves aren't predictors.

- L319-320; incomplete sentence

- L333-335; "(increases of trends)" not needed as it is explained what is meant by break-points in the second part of this sentence.

---

## Referee Comment (RC2) · Anonymous Referee #2 · 1 Nov 2018

• "Seasonal sea ice prediction based on regional indices" is an intriguing new look into the statistical predictability of sea ice conditions as defined by the Barnett Severity Index (BSI). The BSI is one of the longer duration sea ice metrics on record and I commend the authors for reaching back prior to the satellite era to paint a more thorough picture of sea ice conditions. We need more work in this area. • As the authors note, this paper somewhat extends previous work by Drobot but it is not a precise analogue. The present paper has three main objectives: (1) to quantify predictability inherent in antecedent spatial distributions of sea ice, (2) to distinguish predictability of pan-Arctic sea ice from that of regional predictability, and (3) to distinguish quantitatively the trend-derived predictability and predictability of departures from trend. • What is written in the paper is very well done. I have few critical issues with this aspect

of the paper. Assessing the predictability of de-trended data is a key baseline contribution for further developing statistical sea ice forecasts. However, I am left wanting with just this analysis. In the Drobot paper, sea ice data was supplemented with atmospheric teleconnections and other data. This paper would be greatly strengthened by adding additional predictors so we can begin to better understand the predictability of de-trended sea ice data. • For the above reason, I recommend this paper needs major revisions. To clarify, I find little fault with what is here – it's just that I don't think it's enough. By adding additional predictors, this will become a more complete package and one that will have high visibility moving forward.
* * *

---

## Author Comment (AC1) · 30 Nov 2018

Responses to Reviewers

First of all, we wish to thank both reviewers for their careful reading of the paper and for the insightful and constructive suggestions. In the following pages, we provide point-by-point responses following every comment from the reviewer.

Anonymous Referee #1

Overview and broad comments

This study by Walsh et al uses a long historical record of sea ice coverage data to estimate the amount of variance explained in two (mainly September) quantities, the

Beaufort Sea Index and pan-Arctic sea ice extent, by the sea ice extent in different Arctic sub-regions. They consider both concurrent correlations and lagged correlations of these quantities with sea ice conditions for previous months, and separate these correlations into a total and interannual estimation. They argue that a piece-wise linear trend is most appropriate for detrending the different time series for estimating the latter.

The noteworthy pieces of information to come out of this study are (1) a piece-wise linear trend is the best option for detrending the time series considered in the study, (2) the break-point year emergent in that analysis which can be interpreted as an acceleration of the ongoing negative trend in sea ice cover is in the mid- to late- 1990's, (3) that interannual variability in the BSI can be explained by June sea ice coverage in the Beaufort Sea and by July coverage in the Chukchi Sea, (4) consistent with other studies the September pan-Arctic ice extent has significant autocorrelation back to July (about two months), and (5) that the Laptev and East Siberian Seas explain the most concurrent correlation in September pan-Arctic SIE.

As it stands though, in my opinion the paper requires substantial revisions and additional analyses before being re-submitted.

The key goal of the study seems to be to provide baseline metrics against which sea ice prediction studies can be evaluated. However, that baseline has already been established for one of the two predictands considered in this study, pan-Arctic SIE, in several other studies based on autocorrelation (i.e. persistence). No reference to these other studies is ever made. The one thing that separates the result about the pan-Arctic SIE predictand in this study from others is that a very long historical record was used here, but that point should be emphasized when motivating the study (it's currently not mentioned). The rest of the lagged correlation analysis for pan-Arctic SIE, which was between it and the SIE for the various subregions, didn't yield higher correlations than lagged pan-Arctic sea ice extent itself. So, for evaluating prediction skill, is it not the autocorrelation of pan-Arctic SIE that is the important baseline to

beat? What therefore is the motivation to consider the lagged correlation analysis with the different sub-regions?

We agree that the baseline for persistence-based predictions have been established in previous studies (e.g., Blanchard-Wrigglesworth et al., 2011; Day et al., 2014; Bushuk et al., 2017, 2018), and the revision places added emphasis on the use of a longer record length (back to 1953 rather than 1979) in this regard. The use of autocorrelations and cross-correlations was essentially a vehicle for illustrating the issues associated with detrending in a predictive framework. The main intent of the paper is to show how detrending is a key step in the depiction of persistence-based statistical predictions. We illustrate the effect of detrending for both pan-Arctic ice extent and regional metrics in order to show that predictive applications on both scales must address detrrending in a rigorous way, and that there are various alternatives for detrending. While these alternative detrending strategies are known, the relative effectiveness of the various alternatives has not been addressed in previous studies. (Goldstein et al., 2016, The Cryos. Disc.; 2018, Sci. Rep.) come closest by comparing representations based on linear trends and discontinuities in the mean. An addition novel outcome of the present study is the synthesis of break-point information across the various Arctic sub-regions and (in a reviewer-based revision) across seasons.

With respect to the BSI predictand, that analysis seems okay and is fine to report on, but I'm not sure all of the information presented (particularly in the tables; see specific comments below) is needed to reach the conclusions made.

As indicated in the detailed responses below, we are relegating two of the the tables to Supplementary Material.

I was also a bit surprised (and disappointed) to see that only these two predictands were considered if the goal is to provide baseline skill numbers, especially considering the first predictand has already received considerable attention in other studies. Like in the supplementary material of Bushuk et al 2018 (reference below), it would be good

to extend the analysis to treat all of the different regions as separate predictands and compute the autocorrelations for each of those regions. This would help put the results of the Bushuk et al study into context, as they used a shorter and different observational dataset than is used here. As of now, and considering some of my suggestions to remove certain parts of the results section (see specific comments below), this additional analysis would help strengthen that section and would provide useful baseline numbers for future studies.

This suggestion is incorporated into the revision, and it is especially helpful in emphasizing that a novel feature is the use of longer time series than have been used on other recent studies that used only the post-1978 satellite data. In the revision, we present comparisons of the regional autocorrelations for two periods, 1953 onward and 1979 onward, with explicit reference to the Bushuk et al. (2018) regional autocorrelations.

I was missing from the introduction a sufficient argument for the need for this study in the context of current literature on sea ice prediction. The additional references to, and discussion of some key studies listed below are needed to place the goals of this study and its findings into perspective. Specifically here are studies on persistence/autocorrelation of observed and modeled pan-Arctic and regional sea ice extent:

Blanchard-Wrigglesworth, Edward, et al. "Persistence and inherent predictability of Arctic sea ice in a GCM ensemble and observations." Journal of Climate 24.1 (2011): 231-250. – Includes estimates of autocorrelation for pan-Arctic SIE and SIA in both observations and models. Also includes contributions of lagged Beaufort Sea sea ice thickness to september SIA and SIE variability. – This paper was referenced in the study, but it's odd that only the SST re-emergence result was mentioned.

Sigmond, M., et al. "Seasonal forecast skill of Arctic sea ice area in a dynamical forecast system." Geophysical Research Letters 40.3 (2013): 529-534. – Their Fig. 3 shows persistence forecasts of SIE from two observational datasets.

Bushuk, Mitchell, et al. "Regional Arctic sea–ice prediction: potential versus operational seasonal forecast skill." Climate Dynamics (2018): 1-23. − In their supplementary material, they show autocorrelations for individual regions from observations and compare against models.

These references are indeed relevant. We have carefully gone through each one and identified points of comparison for inclusion (with references) in the revision. As noted above, we also include quantitative comparisons of the post-1953 and post-1979 autocorrelations in order to broaden the temporal context relative to the previous studies. We distinguish between the persistence information derived from models and from observational data, as studies such as Blanchard-Wrigglesworth et al. (2011), Sigmund et al. (2013) and Bushuk et al. (2018) include both. We also add information (including explicit references) to other relevant papers that have addressed trends and autocorrelation issues in a predictive context: Day et al. (2014, J. Climate), Bushuk et al. (2017, J. GRL), Goldstein et al. (2016, The Cryos. Disc.).

Throughout the paper, the word "predictability" is used somewhat misleadingly. It should be made clear early on that what is being referred to as predictability throughout the rest of the paper is a very specific estimate of predictability – that is, the predictability of a certain sea ice coverage quantity based on the lagged cross-correlation or auto-correlation (often referred to as memory or persistence) between that quantity and other sea ice coverage quantities. Otherwise, it sounds like the authors are referring to predictability in general, which encompasses all sources of predictability and estimates of theoretical predictability limits.

Agreed. We have drafted a paragraph, to be inserted following line 82, clarifying that we are addressing only persistence-based statistical predictability based on the observational record, and we will state clearly that there are other sources of predictability, especially those captured by physical-dynamical models of the coupled atmosphere-ocean-ice system. We will insert persistence-based in front of predictability, beginning with line 12 in the abstract. We do note that the original manuscript was "up front" in referring to "forecast skill achieved by other methods such as more sophisticated

statistical formulations, numerical models, and heuristic approaches" (Abstract, first sentence).

Guemas, Virginie, et al. "A review on Arctic seaice predictability and prediction on seasonal to decadal timescales". Quarterly Journal of the Royal Meteorological Society 142.695 (2016): 546-561. – Provides a review on sources of predictability, including memory/persistence of sea ice coverage and thickness. See references therein.

Bushuk, Mitchell, et al. "Regional Arctic sea–ice prediction: potential versus operational seasonal forecast skill." Climate Dynamics (2018): 1-23. – Focuses on regional predictability.

The revised Introduction summaries the conclusions in Guemas et al. (2016) concerning contributions of the atmosphere and ocean to sea ice predictability, and it highlights Bushuk et al.'s (2017) finding that the ocean surface and subsurface state contributes to predictability in the North Atlantic subarctic.

The de-trending analysis could be expanded on. The 1979-onward period is what is commonly used in prediction studies, and there is an ongoing debate in the literature about what de-trending method is most appropriate for pan-Arctic SIE, particularly when the most recent few years are included. Extending the de-trending analysis to focus also on this shorter period would be helpful for future prediction studies. While most authors choose to fit a linear trend over that period, Dirkson et al 2017 suggested a quadratic fit; others suggest de-trending with a high-pass filter but this has the unfortunate effect of removing the first or last sample from the data over an already short period. It would be worthwhile, and would add to the paper, if the authors could make a case for the piece-wise linear trend over this shorter period if indeed it is much better than a linear or quadratic fit. Also, as prediction studies are beginning to focus more on regional sea ice prediction than on pan-Arctic SIE/SIA, it would be helpful to know what choice is most appropriate for the different regions and whether this choice depends on the month or season. For instance, Bushuk et al 2017 and Dirkson et al 2017 ar-

gued that linear de-trending was sufficient for regional SIE and local SIC over a similar period considered here.

Dirkson, Arlan, William J. Merryfield, and Adam Monahan. "Impacts of sea ice thickness initialization on seasonal Arctic sea ice predictions." Journal of Climate 30.3 (2017): 1001-1017.

Bushuk, Mitchell, et al. "Skillful regional prediction of Arctic sea ice on seasonal timescales." Geophysical Research Letters 44.10 (2017): 4953-4964.

Following this suggestion, we have expanded on our evaluation of the benefits of a piecewise linear fit relative to a single linear trend line. Specifically, we have identified all cases (across regions and the winter, spring and summer seasons) in which the piecewise fit reduces the residual variance by more than 5% relative to a single linear trend. Only these cases are included in our (revised) summary figure containing – for each season separately – the temporal histograms of the break-point years. The revised figure, containing plots for the three seasons preceding September, will replace the original Figure 5, which combined the results for the three seasons into a single bar plot.

Specific Comments

 ć It should be stated in the abstract early on (before L11-13) the prediction method that will be used in this paper.

The text beginning on line 11 of the Abstract is modified to: "In this study, we use observational data to evaluate the contribution of the trend to the skill of persistence-based statistical forecasts of monthly and seasonal ice extent on the pan-Arctic and regional scales."

 ć L19: "statistical predictability" should be replaced with "statistical skill..." based on said approach. It's not statistical predictability in the sense of a theoretical limit, which isn't directly possible to estimate.

Agreed. Will use "statistical skill" here, and in other instances where "predicatability" had been incorrectly used. The previous item (revision of lines 9-11 in the Abstract) is an example of the rewording to "skill" rather than "predicatability".

• L17-18; Which month(s) are being referred to here?

Revised to say "September trend".

• L31-32; Statement requires a reference.

The AMAP (2017) reference in the preceding sentence will be inserted here. Can also add a reference to a new paper on Arctic indicators: Box et al. (2018, "Key indicators of Arctic climate change, 1971-2017", Env. Res. Lett., in press).

• L37-39; Should cite: Maslanik, James, et al. "Distribution and trends in Arctic sea ice age through spring 2011." Geophysical Research Letters 38.13 (2011).

Revised to cite Maslanik et al. (2011) and also the more recent AMAP (2017).

• L47-50; This positive trend in the Bering Sea in winter is disappearing if more recent years are taken into account... Ah, this is stated in the conclusions, but is probably more appropriate to place here instead.

New statement added at end of paragraph (line 51): "The only region in which sea ice shows a weak positive trend has been the Bering Sea (Parkinson, 2014), although the extreme negative sea ice anomalies in the Bering Sea during the past few years have essentially eliminated any positive trend."

• The departures are also affected by antecedent sea ice conditions themselves. Please see the references given in the general comments above related to persistence.

We assume this comment refers to lines 74-76, where we will add ""in addition to antecedent sea ice conditions themselves" to the sentence ending on line 76.

• L79-82; This is not correct. "Ice-ocean model" implies that both the ice and ocean

evolve freely as determined by the model (only the atmospheric forcing is re- quired). Also, the statement ignores the fact that fully coupled models, which de- termine both the atmospheric and ocean/ice conditions prognostically, are now used more often for sea ice prediction. These models are also limited by the chaotic nature of the climate system, but this is typically accounted for by running ensembles. There is important literature on predictability using said models that isn't referenced or discussed here. Please see general comment above highlighting predictability literature.

Text on on lines 79-80 is modified as follows: "Even ice-ocean models, which are ini- tialized to current sea ice and ocean conditions, require atmospheric forcing in order to predict future ocean states. Moreover, fully coupled models, which determine both the atmospheric and ocean/ice conditions prognostically, are now used increasing often for sea ice predictions. Ensembles of coupled simulations are generally run because of the chaotic nature of the climate system. These models can be run for much longer time periods than the observational sea ice record, so they can provide statistics of sea ice persistence (autocorrelations) subject to the "perfect model" assumption. Ex- amples of studies employing the "perfect model" approach are Holland et al. (2010. Clim. Dyn.), Blanchard-Wrigglesworth et al. (2011), Day et al. (2014), Bushuk et al. (2018) and Bushuk et al. (2018). In these model simulations, autocorrelation of sea ice anomalies tends to be greater in the model results than in observational data (e.g., Blanchard-Wrigglesworth et al., 2011, their Fig. 2; Day et al., 2014, their Fig. 1)."

• L83-84; It's difficult to see how this paper is an "extension" of Drobot 2003; while this study also focuses on the Beaufort and Chukchi Seas, and extends an analy- sis of statistical prediction skill in that region based on more recent observational data, the analysis done in Drobot 2003 (statistical prediction using multiple linear regression with many more predictors) is not repeated here.

We will delete this sentence, but will retain the subsequent summary (lines 84-92; 95-96) of Drobot's (2003) relevant study.

• L88-92; The caution raised in the Drobot 2003 study I think definitely deserves attention given Arctic sea ice and Arctic climate have changed, as stated. Al- though this study distinguishes between prediction of sea ice coverage with and without the trend, as written it sounds like the authors will carry out an analysis similar to Drobot 2003, and determine the impact of the changing conditions on the statistical relationships found in that study, which is not what is done here.

Lines 91-92 revised to "While the present study will not include the type of multiple-predictor evaluation carried out by Drobot (2003), it will provide a more regionally comprehensive and updated assessment of sea ice anomaly persistence in a predictive context".

• L95-96; This is incorrect. While the Drobot 2003 study didn't consider the effects of detrending as stated, Blanchard-Wrigglesworth et al 2011 did. Specifically they detrended the model projections of SIE by subtracting the ensemble mean at each point in time (removing the forced signal). Additionally, they detrended the SIE observations by subtracting the long-term linear trend.

Sentence in lines 95-96 will be deleted.

• L97-100; I find this overview too vague and is broader in scope than what is actually done in the paper. I think it would be more accurate and helpful to the reader if the authors provided, in order of appearance in the paper, what will specifically be done.

Lines 97-100 will be replaced by the following paragraph containing a more specific summary of the following sections: "In the present paper, we use the autocorrelation statistic to quantify the skill of persistence as a control forecast of pan-Arctic and regional sea ice extent. In addition to utilizing the more conventional metric of ice extent in regional and pan-Arctic domains, we include a regional sea ice index developed in the 1970s to capture interannual variations of marine access in the Beaufort Sea.. A primary focus of the evaluation is the method of detrending the data, as various alternative methods have not been fully explored in the literature. We show that the piecewise

linear method generally results in the smallest residual variance about the trend line, and we then perform an across-region synthesis of information on the break-points in the trend lines in different seasons. Our period of analysis extends back to 1953, which results in a considerably larger sample of years than the more commonly used satellite period (1979 onward). Finally, we examine lagged cross-correlations to determine whether pan-Arctic ice extent or Beaufort Sea summer ice conditions are foreshadowed in a statistical sense by antecedent ice conditions in particular subregions of the Arctic."

• L102; "seasonal climatologies, persistence, and trend" of what? Should clarify that you are referring to these quantities for sea ice coverage itself.

Revised to "arising from climatological sea ice coverage, sea ice persistence, and sea ice trends".

• L104; "ice-ocean models" should be replaced with "dynamical models". Ice- ocean models are a specific subset of these, but fully-coupled models are used more often.

Revised to "dynamical models" to include fully coupled models as well as ice-ocean models.

• L105; The Sea Ice Outlook was managed by the Sea Ice Prediction Network, and starting in 2018 became managed by the Sea Ice Prediction Network–Phase 2. The Sea Ice Outlook is not a previous name for Sea Ice Prediction Network. See https://www.arcus.org/sipn/sea-ice-outlook.

Revised to "The Sea Ice Outlook, coordinated by the Sea Ice Prediction Network now in its Phase 2 (https://www.arcus.org/sipn/sea-ice-outlook), provides an annual compilation. . ."

• L118; Suggest changing "maps" to "gridded records". Presumably what is being referred to here is data in digitized form and not the presentation of the data literally as maps.

Will change "maps" to "digitized records"

• L124-127; Say the product name here. The product is described in the next sentence without actually saying explicitly that it is used.

Revised to "...we compute ice extent using the gridded Arctic-wide sea ice concentration product known as "Gridded Monthly Sea Ice Extent and Concentration, 1850 Onward (Walsh et al., 2015), referred to in the National Snow and Ice Data Center (NSIDC) catalog as G10010. This dataset is based on observations from approximately 15 historical sources...".

• L154-155; While this is true, previous studies have shown that there are shorter persistence time scales for pan-Arctic SIE than for pan-Arctic SIA due to high frequency dynamical influences that change SIE, but not SIA (Blanchard- Wrigglesworth et al 2011; their Fig. 14). This should be mentioned here.

Will insert the following after line 156: "It should be noted, however, that persistence time-scales of pan-Arctic sea ice area have been shown in previous studies (e.g., Blanchard-Wrigglesworth et al., 2011) to be longer than those of pan-Arctic sea ice extent because high-frequency forcing can change ice extent more than it changes ice area (i.e., by converging or diverging ice floes in the absence of ridging or melt)."

• L157ff; Was there any interpolation done to get the G10010 dataset onto the same grid used to define the MASIE sub-regions?

No. The MASIE region mask defines the regions on its grid. For our comparisons, we used the region code of the closest MASIE grid cell for each grid cell in the G10010 grid. No data needed to be interpolated to re-project the MASIE region mask onto the G10010 grid.

• L182; Somewhere early on in the methods section it should be stated what years are considered in the study.

Will change the first sentence of Section 3 to say "As shown in Figure 3, Arctic ice

extents have generally been decreasing over the post-1953 period of this study."

â˘á L191-192; Please clarify. I think what is meant is that the presence of a trend in a time series inflates forecast skill when using the anomaly correlation coefficient to assess that skill. It should be noted though that for distance metrics like the root mean square error, skill can actually be inflated by detrending the data if the two time series have different magnitudes of trends.

This statement referred to the use of simple persistence (autocorrelation) of the time series to be predicted, in which case there is no second time series. However, we see the reviewer's point for cases in which a second variable (e.g., another region's ice extent) is the predictor. The reword Lines 191-192: "However, a trend can inflate persistence-based forecast skill when a variable is used to predict itself (assuming the historical trend continues into the future)".

â˘á L198-202; please refer to general comment on de-trending, but this paragraph would be a good place to reference choices made in other sea ice prediction studies.

While the revision includes new references and descriptions of their relevance to the present study (cf. response to this reviewer's earlier paragraph beginning with "The key goal of this study..."), we will also replace the sentence in Lines 202-204 with the following summary: "The previous studies cited earlier (e.g., Blanchard-Wrigglesworth et al., 2011; Sigmund et al., 2013; Day et al., 2014; Bushuk et al., 2017, 2017) have generally relied on least-squares linear fits for detrending. Goldstein et al. (2016, 2018), by contrast, showed that discontinuous changes in the mean better captured time series (such as open water area) characterized by abrupt changes. In the spirit of the Goldstein et al. studies, we explore various options for detrending a time series such as those in Figures 2 and 3, for which the changes are more pronounced in recent decades than in earlier decades. In such cases, a single multi-decadal trend line cannot be expected to optimally represent the historical evolution."

â˘á L209ff; As of now, I find the description of the fitting methodology a bit hard to

[Figure]

Interactive
comment

follow. It could be more straight-forward both in terms of its description and implementation. If one thinks of the function as a piece-wise linear trend defined by

y=a1x+b1, x < xb (1) and y=a2x+b2, x > xb (2)

where xb is the breakpoint year, continuity of y at xb can be ensured by letting b2= (a1−a2)xb+b1. Substituting b2 into the equation yields 4 parameters to be estimated (a1, b1, a2,xb), which can be done using the 'curve_fit' function. This avoids having to also use both the 'curve_fit' and 'lin_regress' function, as the parameters found by 'curve_fit' can be used to describe the two lines completely. Regardless of which method is used, the equation(s) used in 'curve_fit' should be shown in the paper.

The procedure outlined by the reviewer essentially captures the procedure we used. The "curve_fit" function is defined in lines 504-794 of the file https://github.com/scipy/scipy/blob/master/scipy/optimize/minpack.py, which we will add to the description of our methodology after Line 216. This function performs a least-squares fit to the function by modifying the function's parameters. A starting "guess" of the function parameters is provided by the user. The linear algebra methods of the scipy numerical library is then used. The slopes and break points that emerge will be the same as those obtained by the use of (1) and (2). .

• L227-229; This was just stated on L207-208.

There is a slightly different message in the two statements, with the first (Lines 207-208) referring to single time series while the second (Lines 227-229) refers the across-region comparison of the break points. The revision clarifies this distinction by changing Lines 227-229 to: "Because the break-points are computed separately for each region, the use of the two-piece linear fit allows comparisons of the timing of the break-points across the various subregions".

• L239ff; This is an interesting result, but is it necessary to include the break-point year for all calendar months from January-September in Figure 5? The piece-wise

equation seems like a sensible approach to determine a breakpoint year in the summer months (except maybe in the central Arctic). However, for winter and spring months is it really the case that a two-piece linear trend yields a better mean squared error than a simple linear trend? If not, then the breakpoint year for those months won't carry any significant meaning other than that it's an emergent parameter you get from fitting the two-piece linear trend. Is that perhaps what is being seen in Fig. 5 for years before the 1990's and after the early 2000's? Breaking down the information in Fig. 5 for each month and region would make the results easier to interpret and more informative.

The reviewer raises an important point concerning the seasonality of the break-points. In response, we have expanded this section by constructing histograms of break-points for the three antecedent seasons: Jan-Mar, Apr-Jun, and Jul-Sep. The revised Figure 5 now contains three panels, one for each season. (We limited the break-points included in the plots to those for which the two-piece fit reduced the variance about the trend lines by at least 5% relative to a single linear trend line). The result is that the clustering of break-points in the 1990s is more apparent in summer and winter than in spring, although the number of plots satisfying the variance-reduction criterion is greater in the summer than in the winter.

• L250; Is the reason for showing Figure 6 not also to see what regions contribute the most explained variance to pan-Arctic SIE?

Yes. Line 250 is revised to "In order to illustrate the effect of detrending and to show which regions contribute the most explained variance to pan-Arctic sea ice extent, Figure 6 shows. . .".

• L256-260; Why say the significance level for this specific autocorrelation value? Is it the maximum autocorrelation value for all of the samples considered?

We are not sure which "specific autocorrelation" value the reviewer is referring to, as lines 255-256 provide the ranges of our correlations. The confusion may arise from the significance thresholds on lines 256-258, for which we also provide a range that encompasses the corresponding range of autocorrelations (from 0 to 0.40). We will insert a statement in Line 258 that "None of the regional or pan-Arctic ice extent correlations exceeded 0.40)."

• L260-262; This is the significance value when the sample has an autocorrelation of zero. Do the detrended time series have no autocorrelation? Also, this statement is true for most regions, but not all according to Fig. 6.

Figure 6 shows cross-correlations between regional and pan-Arctic ice extent, not auto-correlations. The detrended time series have very small autocorrelations (0.15 or less), which results in effective sample sizes greater than 50 (following Bretherton, 1999), in which case the 95% significance threshold increases only from 0.26 to 0.28 – with no changes to our conclusions about statistical significance.

• L263-265; This is true when the trend is included, but not when the time series are detrended according to Fig 6. Please clarify which is being referred to here. Can the authors speculate at all why it is the East Siberian Sea and Laptev Sea explain the most synchronous interannual variance in pan-Arctic SIE? Is it the region that has the most interannual variability in September?

Yes, this statement needed to be revised for clarification, as noted by the reviewer. To be precise: "According to Figure 6, the regions contributing most strongly to September pan-Arctic sea ice variations (including trends) are the Beaufort, Chukchi and East Siberian Seas. After the data are detrended, the regions contributing most to September pan-Arctic sea ice variations are the East Siberian and Laptev Seas". The somewhat surprisingly large contribution of the Laptev Sea is consistent with the "dynamical preconditioning" hypothesis of Williams et al. (2016), which we cite in the revision. The variances of the detrended September extents of East Siberian and Laptev Seas are indeed among the largest of all the regions, although the Chukchi Sea's interannual variance is essentially as large".

• L266-275; Is Figure 7 really necessary? It seems reasonable to consider the

contributions of variability in different regions to variability in pan-Arctic SIE, but is there any physical basis for thinking that regions far away from the Beaufort Sea would show any explained variance in the BSI, apart from the trend? It's worthwhile to know how far from the Beaufort Sea variations matter, but those regions are shown in Fig. 9 already for lag 0.

While we are open to removing material (see following comment), we would prefer to retain Figure 7 if possible because it shows that regions of significant explained variance include the Canadian Archipelago to the east, as well as the Chukchi Sea to the west. There is a difference in the "scale of influence" in Figures 6 and 7 that is worth noting, and we will add a statement to that effect (at Line 275).

• Are tables 1-4 necessary?

No, the tables are not necessary except for supporting the finding that a larger fraction of September pan-Arctic variance is explained by antecedent pan-Arctic extent than by antecedent regional extent (see comment below on Figure 8). If the journal allows Supplementary Material, we will relegate Tables 1-4 to the SM; otherwise we will omit Tables 3 and 4, but would prefer to retain Tables 1 and 2 to support the above finding. The text in Lines 276-292 will be modified to summarize the results presently therein without citing Tables 3 and 4 unless those tables are included as SM.

For Tables 1 and 3, when the full time series are compared (non-detrended), the fact that there will be correlation between the different regions and the 2 predictands when both predictor and predictand contain trends is not really a surprise, is it? For Tables 2 and 4, the only relevant information to prediction of the BSI is limited to a few (expectantly nearby) regions, and for the most part to a short number of lag months. For those areas and lags where there is any explained variance over 10% , that information is already plotted in Figs. 8 and 9. Why not just say when describing Figs 8 and 9 that no explained variance values greater than 10% were found in other regions?

It is not surprising that the trends inflate the correlations, although the extent to which

the correlations decrease with detrending might not have been anticipated without actually evaluating the cross-correlations. There is indeed a limited spatial scale of coherence in the BSI results in Tables 2 and 4. We will follow the reviewer's suggestion and shorten the text by deleting Lines 314-320 and replacing it with "The BSI variance explained by all other regions is less than 10%".

If an argument to keep the tables is made, the shading scheme should be explained.

As noted above, Tables 3 and 4 will be deleted from the main text. The two levels of shading simply denote explained variances that exceed 10% and 40%.

• L284ff and Figure 8; I'm surprised to see the values of 0.41 and 0.05 for the Beaufort Sea in Fig. 8 (and identical values for the Canadian Archipelago in the tables 1, 2) for lag months January-April. How can this be if sea ice extent is not variable in the Beaufort Sea and the Canadian Archipelago in the winter?

These values are actually misleading, and they are consequences of very small digitization artifacts. In one of the earlier sources of the pre-satellite era, the winter sea ice concentrations were digitized with a slightly different land mask. This biased digitization impacted both the regional (Beaufort) and Canadian Archipelago) as well as the pan-Arctic ice extents. The revised will contain correlations based on corrected values, and the revised correlations will be zero.

• Figure 8; I think it makes sense to also show the autocorrelation for September pan-Arctic SIE (recognizing it is in Tables 1 and 2; see comment above). It would put the contributions of these other regions into context for prediction purposes. According to Table 2, Pan-Arctic SIE contributes at different lags contributes more to September SIE than any individual region. Should that result itself be mentioned as the most significant (albeit arguably expected) result in terms of prediction? This result should be compared against the studies mentioned in the general comments, but it seems to agree with them. Also, the lagged Laptev Sea SIE contributes more to September Pan-Arctic SIE than the Beaufort Sea, so why not show it too?

As suggested, we are adding pan-Arctic and Laptev Sea panels to Figure 8. The revised text also highlights the result that the pan-Arctic extent of July and August indeed correlates more highly than any regional extent with September pan-Arctic ice extent in both the non-detrended and the detrended data (Tables 1 and 2). This is indeed the rationale for retaining Tables 1 and 2. The finding that the lagged pan-Arctic correlations exceed the lagged regional vs. pan-Arctic correlations is consistent with the perfect-model results in Bushuk et al.'s (2017) Figure 2, which we will cite, although this comparison is not apples-vs.-apples: Bushuk et al. show the skill pf predictions of regional extent (not pan-Arctic extent) in their regional panels. The same is true for Day et al.'s (2014) Fig. 11 and for Bushuk et al.'s (2018) Figs. 6/9/10/11.

A physical explanation for Laptev Sea can be found in Williams et al 2016: Williams, James, et al. "Dynamic preconditioning of the minimum September sea-ice extent." Journal of Climate 29.16 (2016): 5879-5891.

The revision will cite this paper and its explanation (cf. response to comment on Lines 263-265).

 c L288-291; "...reaching zero by...". Near zero, not zero exactly. It's also odd that these are not zero exactly when ice covers these regions completely through April

Revision of Line 289: "…approaching zero by 3-4 months". The non-zero values are another consequence of the land-mask difference noted above (cf. response to comment on L284 and ff), and the revision will contain correlations based on corrected (temporally consistent) values of ice extent.

 c L319-320; Doesn't the BSI contain information from earlier months than September? That is, it's not a strictly September prediction metric.

Yes, the BSI includes pre-September information such as the length of the navigation season in the Beaufort. In the revision, we add the following to Line 320: "This percentage of explained variance is even less than one might have anticipated, given that

the BSI includes information on the length of the navigation season, which can begin well before September, i.e., as early as July in some years".

• L336-338; Specify that this is a general finding; it's certainly not true for all "ice extents" considered.

Revision of Line 336: "Based on the raw (not detrended) time series, the antecedent ice extents in a substantial fraction of the Arctic regional seas provide significant predictive skill...".

• L338-340; "(and even years, given the multidecadal scales of the trends)"... While probably true, this isn't a direct conclusion/finding of this study.

Will delete the parenthetical phrase, so the revised statement is limited to "...the regional extents of prior seasons".

• 342-345; The study being referenced here is based on six years of hindcasts which were created by averaging over many different techniques/models, and the results are not reflective of the predictive skill shown by several other prediction studies. To put the results of this paper into context, it should be stated that numerous other studies have shown higher skill than anomaly persistence forecasts (which is essentially the method used here). See references in the introduction of Bushuk et al 2018 (reference in general comments).

We will grant that other sea ice prediction efforts have outperformed persistence and will state this explicitly, citing Bushuk et al. (2018) and studies such as Tivy et al. (2011), Shroeder et al. (2014), Yuan et al. (2016) and Petty et al. (2017). However, in this context, perfect-model studies do not seem to fit the point being made; persistence-derived predictability is greater in perfect models than in corresponding operational forecasts, as even some of the perfect-model studies show. In this respect, the SIPN is the acid test of the current state of sea ice prediction (at least for September pan-Arctic ice extent). A compilation of SIPN results for the past eight years (the "state of the art"),

completed just in the past two weeks by Larry Hamilton, shows that, on balance, the SIPN consensus forecasts outperform detrended anomaly persistence by only a small amount. While that persistence metric is based on year-to-year September variations, the SIPN forecasts for September are made in June, July and August – less than a season ahead, and on the favorable side of the springtime "prediction barrier". So, while the revised text will affirm the reviewer's point, there are some caveats to the claim that numerous studies have shown higher skill than anomaly persistence.

• L355; The influence of trend predictive skill is well recognized in the sea ice prediction community, and skill results based on detrending are commonly presented in sea ice prediction studies, so I'm not sure I see how challenge (1) follows from this particular study.

The reviewer makes a valid point here. The revised text rephrases this "challenge" so that it is merely a call to make optimum use of persistence as a control forecast. In that respect, we mean going beyond simple period-of-record linear trends in in deriving persistence forecasts and in defining anomalies relative to trends.

Technical corrections

• Figure 3: I think it would be easier to see and compare magnitudes of the anomalies and trends if Fig 3 were split into two sub-panels: one for the Beaufort and one for pan-Arctic SIE. The sub-panel for pan-Arctic SIE could have double y-axis (one for March, one for September) so that there isn't such a large vertical space between lines.

Agreed. We have remade Figure 3 into two panels as suggested.

• L194-197; Sentence is worded awkwardly, specifically the "Because..., so ...".

Revision into two sentences: "One of our main interests in this study is whether or not interannual variations of preceding regional ice extents correlate with later BSI values. In order to exclude the effect of the overall trends in the correlation of these time series, we detrend the data and explore various methods for doing so".
 c Figure 4: It looks like there is a kink in the piece-wise linear functions near the breakpoint. This shouldn't be there.

Yes, we see that in the lower panel and have corrected the two-piece linear trend line. Thank you for catching the glitch.

 c L255; "accrual".. what is accruing here? Why not just say "corresponding to correlations..."

That was a typo. Should have been "actual", but "corresponding" is even better.

 c L257 and 258; typo? "a 60-year samples" should be "a 60-year sample"

Revised by deleting "s" at end of "samples".

 c L296; Change to ..."when sea ice extent" for the Beaufort and East Siberian Seas are predictors. The seas themselves aren't predictors.

You are right. We have revised to say that sea ice extents in those seas are the predictors..

 c L319-320; incomplete sentence

"that" (second word) needed to be deleted.

 c L333-335; "(increases of trends)" not needed as it is explained what is meant by break-points in the second part of the sentence.

"(increases of trends)" deleted in revision.

---

## Author Response (AR1)

**Responses to Reviewers**

First of all, we wish to thank both reviewers for their careful reading of the paper and for the insightful and constructive suggestions. In the following pages, we provide point-by-point responses to every comment from the reviewers. Our responses are in italics; the reviewers' comments are in regular font. Line numbers (LXX-LYY) refer to revised manuscript.

**Anonymous Referee #1**

**Overview and broad comments**
This study by Walsh et al uses a long historical record of sea ice coverage data to estimate the amount of variance explained in two (mainly September) quantities, the Beaufort Sea Index and pan-Arctic sea ice extent, by the sea ice extent in different Arctic sub-regions. They consider both concurrent correlations and lagged correlations of these quantities with sea ice conditions for previous months, and separate these correlations into a total and interannual estimation. They argue that a piece-wise linear trend is most appropriate for detrending the different time series for estimating the latter.

The noteworthy pieces of information to come out of this study are (1) a piece-wise linear trend is the best option for detrending the time series considered in the study, (2) the break-point year emergent in that analysis which can be interpreted as an acceleration of the ongoing negative trend in sea ice cover is in the mid- to late- 1990's, (3) that interannual variability in the BSI can be explained by June sea ice coverage in the Beaufort Sea and by July coverage in the Chukchi Sea, (4) consistent with other studies the September pan-Arctic ice extent has significant autocorrelation back to July (about two months), and (5) that the Laptev and East Siberian Seas explain the most concurrent correlation in September pan-Arctic SIE.

As it stands though, in my opinion the paper requires substantial revisions and additional analyses before being re-submitted.

The key goal of the study seems to be to provide baseline metrics against which sea ice prediction studies can be evaluated. However, that baseline has already been established for one of the two predictands considered in this study, pan-Arctic SIE, in several other studies based on autocorrelation (i.e. persistence). No reference to these other studies is ever made. The one thing that separates the result about the pan-Arctic SIE predictand in this study from others is that a very long historical record was used here, but that point should be emphasized when motivating the study (it's currently not mentioned). The rest of the lagged correlation analysis for pan-Arctic SIE, which was between it and the SIE for the various subregions, didn't yield higher correlations than lagged pan-Arctic sea ice extent itself. So, for evaluating prediction skill, is it not the autocorrelation of pan-Arctic SIE that is the important baseline to beat? What therefore is the motivation to consider the lagged correlation analysis with the different sub-regions?

*We agree that the baseline for persistence-based predictions have been established in previous studies (e.g., Blanchard-Wrigglesworth et al., 2011; Day et al., 2014; Bushuk et al., 2017, 2018), and the revision places added emphasis on the use of a longer record length (back to 1953 rather than 1979), cf. new paragraphs:  L122-134; L161-167; L 316-346, as well as new Fig. 5). The use of autocorrelations and cross-correlations was essentially a vehicle for illustrating the issues associated with detrending in a predictive framework.  The main intent of the paper is to show how detrending is a key step in the depiction of persistence-based statistical predictions. We illustrate the effect of detrending for both pan-Arctic ice exten*t *and regional metrics in order to show that predictive applications on both scales must address detrrending in a rigorous way, and that there are various alternatives for detrending.  While these alternative detrending strategies are known, the relative effectiveness of the various alternatives has not been addressed in previous studies, as we state in the new paragraph on p. 6, L149-167. Goldstein et al., 2016, The Cryos. Disc.; 2018, Sci. Rep.) come closest by comparing representations based on linear trends and discontinuities in the mean. An addition novel outcome of the present study is the synthesis of break-point information across the various Arctic subregions and (in a reviewer-based revision) across seasons.*

With respect to the BSI predictand, that analysis seems okay and is fine to report on, but I'm not sure all of the information presented (particularly in the tables; see specific comments below) is needed to reach the conclusions made.

*As indicated in the detailed responses below, we have relegated two of the tables (previously Tables 3 and 4) to Supplementary Material.*

I was also a bit surprised (and disappointed) to see that only these two predictands were considered if the goal is to provide baseline skill numbers, especially considering the first predictand has already received considerable attention in other studies. Like in the supplementary material of Bushuk et al 2018 (reference below), it would be good to extend the analysis to treat all of the different regions as separate predictands and compute the autocorrelations for each of those regions. This would help put the results of the Bushuk et al study into context, as they used a shorter and different observational dataset than is used here. As of now, and considering some of my suggestions to remove certain parts of the results section (see specific comments below), this additional analysis would help strengthen that section and would provide useful baseline numbers for future studies.

*This suggestion is incorporated into the revision, and it is especially helpful in emphasizing that a novel feature is the use of longer time series than have been used on other recent studies that used only the post-1978 satellite data.  In the revision, we present comparisons of the regional autocorrelations for two periods, 1953 onward and 1979 onward, with explicit reference to the Bushuk et al. (2018) regional autocorrelations (new paragraph L316-346 and new Fig. 5).  Our use of the two time periods leads to the conclusion that the springtime "predictability barrier" in regional forecasts based on persistence of ice extent anomalies is not reduced by the inclusion of several decades of pre-satellite data. We highlight this result in the Abstract (L20-22) and in the Conclusion (Section 6), L475-477.*

I was missing from the introduction a sufficient argument for the need for this study in the context of current literature on sea ice prediction. The additional references to, and discussion of some key studies listed below are needed to place the goals of this study and its findings into perspective. Specifically here are studies on persistence/autocorrelation of observed and modeled pan-Arctic and regional sea ice extent:

Blanchard-Wrigglesworth, Edward, et al. "Persistence and inherent predictability of Arctic sea ice in a GCM ensemble and observations." Journal of Climate 24.1 (2011): 231-250.
– Includes estimates of autocorrelation for pan-Arctic SIE and SIA in both observations and models. Also includes contributions of lagged Beaufort Sea sea ice thickness to september SIA and SIE variability.
– This paper was referenced in the study, but it's odd that only the SST re-emergence result was mentioned.

Sigmond, M., et al. "Seasonal forecast skill of Arctic sea ice area in a dynamical forecast system." Geophysical Research Letters 40.3 (2013): 529-534.
– Their Fig. 3 shows persistence forecasts of SIE from two observational datasets.

Bushuk, Mitchell, et al. "Regional Arctic sea–ice prediction: potential versus operational seasonal forecast skill." Climate Dynamics (2018): 1-23.
– In their supplementary material, they show autocorrelations for individual regions from observations and compare against models.

*These references are indeed relevant. We have carefully gone through each one and identified points of comparison for inclusion (with references) in the revision. These references are not cited at various points in the text, beginning with a new Section 2 (Previous Work), L148-167. As noted above, we also include quantitative comparisons of the post-1953 and post-1979 autocorrelations in order to broaden the temporal context relative to the previous studies. We distinguish between the persistence information derived from models and from observational data, as studies such as Blanchard-Wrigglesworth et al. (2011), Sigmund et al. (2013) and Bushuk et al. (2018) include both. We also add information (including explicit references) to other relevant papers that have addressed trends and autocorrelation issues in a predictive context: Day et al. (2014, J. Climate), Bushuk et al. (2017, J. GRL), Goldstein et al. (2016, The Cryos. Disc.).*

Throughout the paper, the word "predictability" is used somewhat misleadingly. It should be made clear early on that what is being referred to as predictability throughout the rest of the paper is a very specific estimate of predictability – that is, the predictability of a certain sea ice coverage quantity based on the lagged cross-correlation or auto-correlation (often referred to as memory or persistence) between that quantity and other sea ice coverage quantities. Otherwise, it sounds like the authors are referring to predictability in general, which encompasses all sources of predictability and estimates of theoretical predictability limits.

*Agreed. We have included a new paragraph (L122-134) clarifying that we are addressing only persistence-based statistical predictability using the observational record, and we have expanded the introductory discussion (L72-106) so that it states clearly that there are other sources of predictability, especially those captured by physical-dynamical models of the coupled atmosphere-ocean-ice system. We have inserted "persistence-based" in front of forecasts and predictability at various points in the text, beginning with L10 in the Abstract. We do note that the original manuscript was "up front" in referring to "forecast skill achieved by other methods such as more sophisticated statistical formulations, numerical models, and heuristic approaches" (Abstract, first sentence).*

Guemas, Virginie, et al. "A review on Arctic seaice predictability and prediction on seasonal to decadal timescales". Quarterly Journal of the Royal Meteorological Society 142.695 (2016): 546-561.
– Provides a review on sources of predictability, including memory/persistence of sea ice coverage and thickness. See references therein.

Bushuk, Mitchell, et al. "Regional Arctic sea–ice prediction: potential versus operational seasonal forecast skill." Climate Dynamics (2018): 1-23.
– Focuses on regional predictability.

*The revised Introduction cites the review paper by Guemas et al. (2016) on L106, and it highlights Bushuk et al.'s (2017) finding that the ocean surface and subsurface state contributes to predictability in the North Atlantic subarctic (L103-105).*

The de-trending analysis could be expanded on. The 1979-onward period is what is commonly used in prediction studies, and there is an ongoing debate in the literature about what de-trending method is most appropriate for pan-Arctic SIE, particularly when the most recent few years are included. Extending the de-trending analysis to focus also on this shorter period would be helpful for future prediction studies. While most authors choose to fit a linear trend over that period, Dirkson et al 2017 suggested a quadratic fit; others suggest de-trending with a high-pass filter but this has the unfortunate effect of removing the first or last sample from the data over an already short period. It would be worthwhile, and would add to the paper, if the authors could make a case for the piece-wise linear trend over this shorter period if indeed it is much better than a linear or quadratic fit. Also, as prediction studies are beginning to focus more on regional sea ice prediction than on pan-Arctic SIE/SIA, it would be helpful to know what choice is most appropriate for the different regions and whether this choice depends on the month or season. For instance, Bushuk et al 2017 and Dirkson et al 2017 argued that linear de-trending was sufficient for regional SIE and local SIC over a similar period considered here.

Dirkson, Arlan, William J. Merryfield, and Adam Monahan.
"Impacts of sea ice thickness initialization on seasonal Arctic sea ice predictions." Journal of Climate 30.3 (2017): 1001-1017.

Bushuk, Mitchell, et al. "Skillful regional prediction of Arctic sea ice on seasonal timescales." Geophysical Research Letters 44.10 (2017): 4953-4964.

*Following this suggestion, we have expanded on our evaluation of the benefits of a piecewise linear fit relative to a single linear trend line. The new text discussing the expanded results is on L316-346. Specifically, we have identified all cases (across regions and the winter, spring and summer seasons) in which the piecewise fit reduces the residual variance by more than 5% relative to a single linear trend. Only these cases are included in our (revised) summary figure containing – for each season separately – the temporal histograms of the break-point years. The revised figure, containing plots for the three seasons preceding September, replaces the original Figure 5, which combined the results for the three seasons into a single bar plot.*

**Specific Comments**

• It should be stated in the abstract early on (before L11-13) the prediction method that will be used in this paper.

*The text beginning on L9 of the Abstract has been modified to: "In this study, we use observational data to evaluate the contribution of the trend to the skill of persistence-based statistical forecasts of monthly and seasonal ice extent on the pan-Arctic and regional scales."*

• L19: "statistical predictability" should be replaced with "statistical skill..." based on said approach. It's not statistical predictability in the sense of a theoretical limit, which isn't directly possible to estimate.

*Agreed. We now use "statistical skill" here (L16) and in other instances where "predicatability" had been incorrectly used. The previous item (revision of lines 9-11 in the Abstract) is an example of the rewording to "skill" rather than "predicatability".*

• L17-18; Which month(s) are being referred to here?

*"September" is now specifically stated (L18, L22).*

• L31-32; Statement requires a reference.

*The AMAP (2017) reference in the preceding sentence will be inserted here (L32). We have also added a reference on L30 to a new paper on Arctic indicators: Box et al. (2018, "Key indicators of Arctic climate change, 1971-2017", Env. Res. Lett., in press).*

• L37-39; Should cite:
Maslanik, James, et al. "Distribution and trends in Arctic sea ice age through spring 2011." Geophysical Research Letters 38.13 (2011).

*Maslanik et al. (2011) is now cited on L42, and also the more recent NOAA (2018) Arctic Report Card (L44).*

• L47-50; This positive trend in the Bering Sea in winter is disappearing if more recent years are taken into account... Ah, this is stated in the conclusions, but is probably more appropriate to place here instead.

*New statement about the Bering Sea has been added (L48-49): "However, the positive trend of Bering Sea ice largely vanishes when the most recent winters (especially 2017-18) are included." We also mention this again in the Conclusion (L455-457)*

• The departures are also affected by antecedent sea ice conditions themselves. Please see the references given in the general comments above related to persistence.

*We assume this comment refers to lines 74-76 of the original manuscript. We have added ",,,in addition to antecedent sea ice conditions themselves" in the revision (L77-78).*

• L79-82; This is not correct. "Ice-ocean model" implies that both the ice and ocean evolve freely as determined by the model (only the atmospheric forcing is required). Also, the statement ignores the fact that fully coupled models, which determine both the atmospheric and ocean/ice conditions prognostically, are now used more often for sea ice prediction. These models are also limited by the chaotic nature of the climate system, but this is typically accounted for by running ensembles. There is important literature on predictability using said models that isn't referenced or discussed here. Please see general comment above highlighting predictability literature.

*Text has been modified as follows (L80-91): "Even ice-ocean models initialized to current sea ice and ocean conditions require atmospheric forcing in order to predict future ocean states. Moreover, fully coupled models, which determine both the atmospheric and ocean/ice conditions prognostically, are now used increasing often for seasonal sea ice predictions. Ensembles of coupled simulations are generally run because of the chaotic nature of the climate system. These models can be run for much longer time periods than the observational sea ice record, so they can provide statistics of sea ice persistence (autocorrelations) subject to the "perfect model" assumption. Examples of studies employing the "perfect model" approach are Holland et al. (2011) Blanchard-Wrigglesworth et al. (2011), Day et al. (2014), Bushuk et al. (2017) and Bushuk et al. (2018). In these model simulations, autocorrelation of sea ice anomalies tends to be greater in the model results than in observational data (e.g., Blanchard-Wrigglesworth et al., 2011, their Fig. 2; Day et al., 2014, their Fig. 1)."*

• L83-84; It's difficult to see how this paper is an "extension" of Drobot 2003; while this study also focuses on the Beaufort and Chukchi Seas, and extends an analysis of statistical prediction skill in that region based on more recent observational data, the analysis done in Drobot 2003 (statistical prediction using multiple linear regression with many more predictors) is not repeated here.

*We have deleted this sentence, but have retained a subsequent summary of Drobot's (2003) relevant study (L102-117).*

• L88-92; The caution raised in the Drobot 2003 study I think definitely deserves attention given Arctic sea ice and Arctic climate have changed, as stated. Although this study distinguishes between prediction of sea ice coverage with and without the trend, as written it sounds like the authors will carry out an analysis similar to Drobot 2003, and determine the impact of the changing conditions on the statistical relationships found in that study, which is not what is done here.

*Text has been revised (L110-112) to "While the present study will not include the type of multiple-predictor evaluation carried out by Drobot (2003), it will provide a more regionally comprehensive and updated assessment of sea ice anomaly persistence in a predictive context".*

• L95-96; This is incorrect. While the Drobot 2003 study didn't consider the effects of detrending as stated, Blanchard-Wrigglesworth et al 2011 did. Specifically they detrended the model projections of SIE by subtracting the ensemble mean at each point in time (removing the forced signal). Additionally, they detrended the SIE observations by subtracting the long-term linear trend.

*Sentence originally in lines 95-96 has been deleted.*

• L97-100; I find this overview too vague and is broader in scope than what is actually done in the paper. I think it would be more accurate and helpful to the reader if the authors provided, in order of appearance in the paper, what will specifically be done.

*Lines 97-100 has been replaced by the following paragraph (L1223-134) containing a more specific summary of the manuscript's subsequent sections: "In the present paper, we use the autocorrelation statistic to quantify the skill of persistence as a control forecast of pan-Arctic and regional sea ice extent. In addition to utilizing the more conventional metric of ice extent in regional and pan-Arctic domains, we include a regional sea ice index developed in the 1970s to capture interannual variations of marine access in the Beaufort Sea.. A primary focus of the evaluation is the method of detrending the data, as various alternative methods have not been fully explored in the literature. We show that the piecewise linear method generally results in the smallest residual variance about the trend line, and we then perform an across-region synthesis of information on the break-points in the trend lines in different seasons. Our period of analysis extends back to 1953, which results in a considerably larger sample of years than the more commonly used satellite period (1979 onward). Finally, we examine lagged cross-correlations to determine whether pan-Arctic ice extent or Beaufort Sea summer ice conditions are foreshadowed in a statistical sense by antecedent ice conditions in particular subregions of the Arctic."*

• L102; "seasonal climatologies, persistence, and trend" of what? Should clarify that you are referring to these quantities for sea ice coverage itself.

*Revised (L136-137) to "arising from climatological sea ice coverage, sea ice persistence, and sea ice trends".*

• L104; "ice-ocean models" should be replaced with "dynamical models". Iceocean models are a specific subset of these, but fully-coupled models are used more often.

*Revised (L138) to "dynamical models" to include fully coupled models as well as ice-ocean models.*

• L105; The Sea Ice Outlook was managed by the Sea Ice Prediction Network, and starting in 2018 became managed by the Sea Ice Prediction Network–Phase 2. The Sea Ice Outlook is not a previous name for Sea Ice Prediction Network. See https://www.arcus.org/sipn/sea-ice-outlook.

*Revised (L139-141) to "The Sea Ice Outlook, coordinated by the Sea Ice Prediction Network now in its Phase 2 (https://www.arcus.org/sipn/sea-ice-outlook, accessed 27 Dec 2018), provides an annual compilation…"*

• L118; Suggest changing "maps" to "gridded records". Presumably what is being referred to here is data in digitized form and not the presentation of the data literally as maps.

*"maps" changed to "digitized records"(L175)*

• L124-127; Say the product name here. The product is described in the next sentence without actually saying explicitly that it is used.

*Revised (L182-186) to "…we compute ice extent using the gridded Arctic-wide sea ice concentration product known as "Gridded Monthly Sea Ice Extent and Concentration, 1850 Onward (Walsh et al., 2015), referred to in the National Snow and Ice Data Center (NSIDC) catalog as G10010. This dataset is based on observations from approximately 15 historical sources…".*

• L154-155; While this is true, previous studies have shown that there are shorter persistence time scales for pan-Arctic SIE than for pan-Arctic SIA due to high frequency dynamical influences that change SIE, but not SIA (Blanchard-Wrigglesworth et al 2011; their Fig. 14). This should be mentioned here.

*New text inserted in L214-218: "It should be noted, however, that persistence time-scales of pan-Arctic sea ice area have been shown in previous studies (e.g., Blanchard-Wrigglesworth et al., 2011) to be longer than those of pan-Arctic sea ice extent because high-frequency forcing can change ice extent more than it changes ice area (i.e., by converging or diverging ice floes in the absence of ridging or melt)."*

• L157ff; Was there any interpolation done to get the G10010 dataset onto the same grid used to define the MASIE sub-regions?

*No. The MASIE region mask defines the regions on its grid. For our comparisons, we used the region code of the closest MASIE grid cell for each grid cell in the G10010 grid. No data needed to be interpolated to re-project the MASIE region mask onto the G10010 grid.*

• L182; Somewhere early on in the methods section it should be stated what years are considered in the study.

*Text has been modified (L246-247) to say "As shown in Figure 3, Arctic ice extents have generally been decreasing over the post-1953 period of this study."*

• L191-192; Please clarify. I think what is meant is that the presence of a trend in a time series inflates forecast skill when using the anomaly correlation coefficient to assess that skill. It should be noted though that for distance metrics like the root mean square error, skill can actually be inflated by detrending the data if the two time series have different magnitudes of trends.

*This statement referred to the use of simple persistence (autocorrelation) of the time series to be predicted, in which case there is no second time series. However, we see the reviewer's point for cases in which a second variable (e.g., another region's ice extent) is the predictor. The reworded L254-255: "However, a trend can inflate persistence-based forecast skill when a variable is used to predict itself (assuming the historical trend continues into the future)".*

• L198-202; please refer to general comment on de-trending, but this paragraph would be a good place to reference choices made in other sea ice prediction studies.

*While the revision includes new references and descriptions of their relevance to the present study (cf. response to this reviewer's earlier paragraph beginning with "The key goal of this study…"), we have also replaced the original Lines 202-204 with the following (L263-271): "The previous studies cited in Section 2 (e.g., Blanchard-Wrigglesworth et al., 2011; Sigmund et al., 2013; Day et al., 2014; Bushuk et al., 2017, 2017) have generally relied on least-squares linear fits for detrending. Goldstein et al. (2016, 2018), by contrast, showed that discontinuous changes in the mean better captured time series (such as open water area) characterized by abrupt changes. In the spirit of the Goldstein et al. studies, we explore various options for detrending a time series such as those in Figures 2 and 3, for which the changes are more pronounced in recent decades than in earlier decades. In such cases, a single multi-decadal trend line cannot be expected to optimally represent the historical evolution."*

• L209ff; As of now, I find the description of the fitting methodology a bit hard to follow. It could be more straight-forward both in terms of its description and implementation. If one thinks of the function as a piece-wise linear trend defined by

$$y = a_1 x + b_1, \quad x < x_b \qquad (1)$$
and
$$y = a_2 x + b_2, \quad x > x_b \qquad (2)$$

where xb is the breakpoint year, continuity of y at xb can be ensured by letting b2= (a1−a2)xb+b1. Substituting b2 into the equation yields 4 parameters to be estimated (a1, b1, a2,xb), which can be done using the 'curve_fit' function. This avoids having to also use both the 'curve_fit' and 'lin_regress' function, as the parameters found by 'curve_fit' can be used to describe the two lines completely. Regardless of which method is used, the equation(s) used in 'curve_fit' should be shown in the paper.

*The procedure outlined by the reviewer essentially captures the procedure we used. The "curve_fit" function is defined in lines 504-794 of the file [https://github.com/scipy/scipy/blob/master/scipy/optimize/minpack.py](https://github.com/scipy/scipy/blob/master/scipy/optimize/minpack.py), which we have added to the description of our methodology after L285. This function performs a least-squares fit to the function by modifying the function's parameters. A starting "guess" of the function parameters is provided by the user. The linear algebra methods of the scipy numerical library is then used. The slopes and break points that emerge will be the same as those obtained by the use of (1) and (2). .*

• L227-229; This was just stated on L207-208.

*There is a slightly different message in the two statements, with the first (previous Lines 207-208) referring to single time series while the second (previous Lines 227-229) refers the across-region comparison of the break points. The revision clarifies this distinction by changing the text (L304-306) to: "Because the break-points are computed separately for each region, the use of the two-piece linear fit allows comparisons of the timing of the break-points across the various subregions".*

• L239ff; This is an interesting result, but is it necessary to include the break-point year for all calendar months from January-September in Figure 5? The piece-wise equation seems like a sensible approach to determine a breakpoint year in the summer months (except maybe in the central Arctic). However, for winter and spring months is it really the case that a two-piece linear trend yields a better mean squared error than a simple linear trend? If not, then the breakpoint year for those months won't carry any significant meaning other than that it's an emergent parameter you get from fitting the two-piece linear trend. Is that perhaps what is being seen in Fig. 5 for years before the 1990's and after the early 2000's? Breaking down the information in Fig. 5 for each month and region would make the results easier to interpret and more informative.

*The reviewer raises an important point concerning the seasonality of the break-points. In response, we have expanded this section (L347-365) by constructing histograms of break-points for the three antecedent seasons: Jan-Mar, Apr-Jun, and Jul-Sep. The revised Figure 6 (formerly Figure 5) now contains four panels, including one for each of the three antecedent seasons. (We limited the break-points included in the plots to those for which the two-piece fit reduced the variance about the trend lines by at least 5% relative to a single linear trend line). The result is that the clustering of break-points in the 1990s is more apparent in summer and winter than in spring, although the number of plots satisfying the variance-reduction criterion is greater in the summer than in the winter.*

• L250; Is the reason for showing Figure 6 not also to see what regions contribute
the most explained variance to pan-Arctic SIE?

*Yes. The original line 250 is revised (L366-367) to "In order to illustrate the effect of detrending and to show which regions contribute the most explained variance to pan-Arctic sea ice extent, Figure 7 shows…".*

• L256-260; Why say the significance level for this specific autocorrelation value?
Is it the maximum autocorrelation value for all of the samples considered?

*We are not sure which "specific autocorrelation" value the reviewer is referring to, as lines 255-256 provided the ranges of our correlations. The confusion may arise from the significance thresholds on lines 256-258, for which we also provide a range that encompasses the corresponding range of autocorrelations (from 0 to 0.40). We have inserted a statement (L375-376) that "None of the regional or pan-Arctic ice extent correlations exceeded 0.40."*

• L260-262; This is the significance value when the sample has an autocorrelation of zero. Do the detrended time series have no autocorrelation? Also, this statement is true for most regions, but not all according to Fig. 6.

*Figure 7 (formerly Figure 6) shows cross-correlations between regional and pan-Arctic ice extent, not autocorrelations. The detrended time series have very small autocorrelations (0.15 or less), which results in effective sample sizes greater than 50 (following Bretherton, 1999), in which case the 95% significance threshold increases only from 0.26 to 0.28 – with no changes to our conclusions about statistical significance.*

• L263-265; This is true when the trend is included, but not when the time series
are detrended according to Fig 6. Please clarify which is being referred to here.
Can the authors speculate at all why it is the East Siberian Sea and Laptev Sea
explain the most synchronous interannual variance in pan-Arctic SIE? Is it the
region that has the most interannual variability in September?

*Yes, this statement needed to be revised for clarification, as noted by the reviewer. The new text (L381-388) reads as follows: "According to Figure 7, the regions contributing most strongly to September pan-Arctic sea ice variations (including trends) are the Beaufort, Chukchi and East Siberian Seas. After the data are detrended, the regions contributing most to September pan-Arctic sea ice variations are the East Siberian and Laptev Seas". The somewhat surprisingly large contribution of the Laptev Sea is consistent with the "dynamical preconditioning" hypothesis of Williams et al. (2016), which we cite in the revision. The variances of the detrended September extents of East Siberian and Laptev Seas are indeed among the largest of all the regions, although the Chukchi Sea's interannual variance is essentially as large".*

• L266-275; Is Figure 7 really necessary? It seems reasonable to consider the
contributions of variability in different regions to variability in pan-Arctic SIE, but
is there any physical basis for thinking that regions far away from the Beaufort

Sea would show any explained variance in the BSI, apart from the trend? It's worthwhile to know how far from the Beaufort Sea variations matter, but those regions are shown in Fig. 9 already for lag 0.

*While we are open to removing material (see following comment), we would prefer to retain this figure (now Figure 8) if possible because it shows that regions of significant explained variance include the Canadian Archipelago to the east, as well as the Chukchi Sea to the west. There is a difference in the "scale of influence" in Figures 7 and 8 that is worth noting, and we have added a statement to that effect (L397-400).*

• Are tables 1-4 necessary?

*No, the tables are not necessary except for supporting the finding that a larger fraction of September pan-Arctic variance is explained by antecedent pan-Arctic extent than by antecedent regional extent (see comment below on Figure 9, formerly Figure 8). Assuming the journal allows Supplementary Material, we have relegated Tables 3-4 to Supplementary Material (Tables S1 and S2); otherwise we will omit Tables 3 and 4, but would prefer to retain Tables 1 and 2 to support the above finding.*

For Tables 1 and 3, when the full time series are compared (non-detrended), the fact that there will be correlation between the different regions and the 2 predictands when both predictor and predictand contain trends is not really a surprise, is it? For Tables 2 and 4, the only relevant information to prediction of the BSI is limited to a few (expectantly nearby) regions, and for the most part to a short number of lag months. For those areas and lags where there is any explained variance over 10% , that information is already plotted in Figs. 8 and 9. Why not just say when describing Figs 8 and 9 that no explained variance values greater than 10% were found in other regions?

*It is not surprising that the trends inflate the correlations, although the extent to which the correlations decrease with detrending might not have been anticipated without actually evaluating the cross-correlations. There is indeed a limited spatial scale of coherence in the BSI results in Tables 2 and 4 (now Table S2). We have followed the reviewer's suggestion and shorten the text by deleting Lines 314-320 and replacing it with "The BSI variance explained by all other regions is less than 10%" (L447-448)*

If an argument to keep the tables is made, the shading scheme should be explained.

*The two levels of shading simply denote explained variances that exceed 10% , 20%, 30%,… This is now explicitly stated in the captions of the tables.*

• L284ff and Figure 8; I'm surprised to see the values of 0.41 and 0.05 for the Beaufort Sea in Fig. 8 (and identical values for the Canadian Archipelago in the tables 1, 2) for lag months January-April. How can this be if sea ice extent is not variable in the Beaufort Sea and the Canadian Archipelago in the winter?

*These values were actually misleading, and they were consequences of very small digitization artifacts. In one of the earlier sources of the pre-satellite era, the winter sea ice concentrations were digitized with a slightly different land mask. This biased digitization impacted both the regional (Beaufort) and Canadian Archipelago) as well as the pan-Arctic ice extents. We have revised the figures (Figs. 9 and 10) to show correlations based on corrected values. The revised correlations are zero where they should be.*

• Figure 8; I think it makes sense to also show the autocorrelation for September pan-Arctic SIE (recognizing it is in Tables 1 and 2; see comment above). It would put the contributions of these other regions into context for prediction purposes. According to Table 2, Pan-Arctic SIE contributes at different lags contributes more to September SIE than any individual region. Should that result itself be mentioned as the most significant (albeit arguably expected) result in terms of prediction? This result should be compared against the studies mentioned in the general comments, but it seems to agree with them. Also, the lagged Laptev Sea SIE contributes more to September Pan-Arctic SIE than the Beaufort Sea, so why not show it too?

*As suggested, we have added pan-Arctic and Laptev Sea panels to Figure 9. The revised text (L426-433) also highlights the result that the pan-Arctic extent of July and August indeed correlates more highly than any regional extent with September pan-Arctic ice extent in both the non-detrended and the detrended data (Tables 1 and 2). This is indeed the rationale for retaining Tables 1 and 2. The finding that the lagged pan-Arctic correlations exceed the lagged regional vs. pan-Arctic correlations is consistent with the perfect-model results in Bushuk et al.'s (2017) Figure 2, which we now cite, although we note that this comparison is not apples-vs.-apples: Bushuk et al. show the skill pf predictions of regional extent (not pan-Arctic extent) in their regional panels. The same is true for Day et al.'s (2014) Fig. 11 and for Bushuk et al.'s (2018) Figs. 6/9/10/11.*

A physical explanation for Laptev Sea can be found in Williams et al 2016:

Williams, James, et al. "Dynamic preconditioning of the minimum September sea-ice extent." Journal of Climate 29.16 (2016): 5879-5891.

*The revised text cites this paper and its "dynamical preconditioning" hypothesis (L385-388).*

• L288-291; "...reaching zero by...". Near zero, not zero exactly. It's also odd that these are not zero exactly when ice covers these regions completely through April

*Revision of Line 289: "...approaching zero by 3-4 months". The non-zero values are another consequence of the land-mask difference noted above (cf. response to comment on L284ff and Figure 8), and the correlations in the revision are based on corrected (temporally consistent) values of ice extent.*

• L319-320; Doesn't the BSI contain information from earlier months than September? That is, it's not a strictly September prediction metric.

*Yes, the BSI includes pre-September information such as the length of the navigation season in the Beaufort. In the revision, we add the following (L446-449): "This percentage of explained variance is even less than one might have anticipated, given that the BSI includes information on the length of the navigation season, which can begin well before September, i.e., as early as July in some years".*

• L336-338; Specify that this is a general finding; it's certainly not true for all "ice extents" considered.

*Revision of Line 336 (now L466-467): "Based on the raw (not detrended) time series, the antecedent ice extents in a substantial fraction of the Arctic regional seas provide significant predictive skill…".*

• L338-340; "(and even years, given the multidecadal scales of the trends)"... While probably true, this isn't a direct conclusion/finding of this study.

*Will delete the parenthetical phrase, so the revised statement (L468-469) is limited to "…the regional extents of prior seasons".*

• 342-345; The study being referenced here is based on six years of hindcasts which were created by averaging over many different techniques/models, and the results are not reflective of the predictive skill shown by several other prediction studies. To put the results of this paper into context, it should be stated that numerous other studies have shown higher skill than anomaly persistence forecasts (which is essentially the method used here). See references in the introduction of Bushuk et al 2018 (reference in general comments).

*We will grant that other sea ice prediction efforts have outperformed persistence and now state this explicitly (L478-480), citing Bushuk et al. (2018) and studies such as Tivy et al. (2011), Shroeder et al. (2014), Yuan et al. (2016) and Petty et al. (2017). However, in this context, perfect-model studies do not seem to fit the point being made; persistence-derived predictability is greater in perfect models than in corresponding operational forecasts, as even some of the perfect-model studies show. In this respect, the SIPN is the acid test of the current state of sea ice prediction (at least for September pan-Arctic ice extent). A compilation of SIPN results for the past eight years (the "state of the art"), completed after the 2018 sea ice minimum, shows that, on balance, the SIPN consensus forecasts outperform detrended anomaly persistence by only a small amount. While that persistence metric is based on year-to-year September variations, the SIPN forecasts for September are made in June, July and August – less than a season ahead, and on the favorable side of the springtime "prediction barrier". So, while the revised text will affirm the reviewer's point, we have added new text (L480-499) placing the improvement over persistence into context.*

• L355; The influence of trend predictive skill is well recognized in the sea ice pre-diction community, and skill results based on detrending are commonly presented in sea ice prediction studies, so I'm not sure I see how challenge (1) follows from this particular study.

*The reviewer makes a valid point here. We have deleted the entire paragraph to which the reviewer refers.*

**Technical corrections**

• Figure 3: I think it would be easier to see and compare magnitudes of the anomalies and trends if Fig 3 were split into two sub-panels: one for the Beaufort and one for pan-Arctic SIE. The sub-panel for pan-Arctic SIE could have double y-axis (one for March, one for September) so that there isn't such a large vertical space between lines.

*Agreed. We have remade Figure 3 into two panels as suggested.*

• L194-197; Sentence is worded awkwardly, specifically the "Because..., so ...".

*Revision into two sentences (L257-260): "One of our main interests in this study is whether or not interannual variations of preceding regional ice extents correlate with later BSI values. In order to exclude the effect of the overall trends in the correlation of these time series, we detrend the data and explore various methods for doing so".*

• Figure 4: It looks like there is a kink in the piece-wise linear functions near the breakpoint. This shouldn't be there.

*Yes, we see that in the lower panel and have corrected the two-piece linear trend line. Thank you for catching the glitch.*

• L255; "accrual".. what is accruing here? Why not just say "corresponding to correlations..."

*That was a typo. Should have been "actual", but "corresponding" is even better. Line 372 has been corrected.*

• L257 and 258; typo? "a 60-year samples" should be "a 60-year sample" (L374)

*Revised by deleting "s" at end of "samples".*

• L296; Change to ..."when sea ice extent" for the Beaufort and East Siberian Seas are predictors. The seas themselves aren't predictors.

*You are right. We have revised to say that sea ice extents in those seas are the predictors (L420)*

• L319-320; incomplete sentence

*"that" (second word) was a typo. Sentence has been revised (L447-450).*

• L333-335; "(increases of trends)" not needed as it is explained what is meant by break-points in the second part of the sentence.

*"(increases of trends)" deleted in revision (L463).*

**Anonymous Referee #2**

˘"Seasonal sea ice prediction based on regional indices" is an intriguing new look into the statistical predictability of sea ice conditions as defined by the Barnett Severity Index (BSI). The BSI is one of the longer duration sea ice metrics on record and I commend the authors for reaching back prior to the satellite era to paint a more thorough picture of sea ice conditions. We need more work in this area.
As the authors note, this paper somewhat extends previous work by Drobot but it is not a precise analogue. The present paper has three main objectives: (1) to quantify predictability inherent in antecedent spatial distributions of sea ice, (2) to distinguish predictability of pan-Arctic sea ice from that of regional predictability, and (3) to distinguish quantitatively the trend-derived predictability and predictability of departures from trend.

What is written in the paper is very well done. I have few critical issues with this aspect of the paper. Assessing the predictability of de-trended data is a key baseline contribution for further developing statistical sea ice forecasts. However, I am left wanting with just this analysis. In the Drobot paper, sea ice data was supplemented with atmospheric teleconnections and other data. This paper would be greatly strengthened by adding additional predictors so we can begin to better understand the predictability of de-trended sea ice data.

*We acknowledge the reviewer's point, which is that the addition of other (atmospheric) predictors, the paper would provide a more comprehensive assessment of statistical predictability. However, previous work by Lindsey et al. (2008, JGR) and Drobot et al. (2006, GRL), has shown that atmospheric modes of variability such as the NAO, AO, and PDO "were found to have little value as predictors of the September Arctic SIE compared with ocean and ice predictors" (Guemas et al., 2014, QJRMS, p. 555). The recent paper by Goldstein et al. (2016, The Cryosphere Disc.), using Self-Organizing Maps (SOMs), also found that there was little evidence of a useful signal of the atmospheric circulation in seasonal prediction of open water season length. Summer wind patterns play a role in interannual variations of late-summer ice extent, but the summer wind patterns in years with similar September pan-Arctic can be quite different (Serreze et al., 2016, JGR) and wind pattern anomalies are largely unpredictable. Ocean anomalies do have some predictive value, especially for the North Atlantic winter ice extent (Bushuk et al., 2017, GRL), but the record of subsurface ocean variables is much shorter than the 60+ year record length of sea ice variations examined in our study. Given that (1) the aggregate of this evidence points to diminishing returns if the predictor suite is expanded beyond sea ice and (2) such an expansion of scope would, in our view, detract from the paper's intentional focus and main "punch lines", we believe that the paper's main messages would be diluted by the expansion to include other types of predictors.*

*Finally, we note that the paper's present focus on persistence generated a host of issues requiring further discussion and clarification, as is evident in the extensive comments of Reviewer 1. While a need for clarification is not a justification for a more limited scope, some of Reviewer 1's comments are fertile ground for further discussion (see Reviewer 1's comment on lines 342-345 and response), so we believe the paper's present content can generate reader interest.*

For the above reason, I recommend this paper needs major revisions. To clarify, I find little fault with what is here – it's just that I don't think it's enough. By adding additional predictors, this will become a more complete package and one that will have high visibility moving forward.

[revised manuscript text omitted]

---

## Author Response (AR2)

**Responses to Reviews of Revised tc-2018-151**

The following are point-by-point responses to the reviewers of the Revised tc-2018-151. The responses are in italics, while the reviewers' comments are in regular font. Line numbers refer to the revised manuscript.

Anonymous Referee #2

**Suggestions for revision or reasons for rejection (will be published if the paper is accepted for final publication)**

Congrats to the author team - the revised manuscript does an excellent job framing the paper and addressing my concerns. I suggest moving to publication.

*We thank the reviewer for the second round of review, and we appreciate the comment. All revisions in this round are in response to Reviewer 1.*

Anonymous Referee #1

Suggestions for revision or reasons for rejection (will be published if the paper is accepted for final publication)

I appreciate the effort made by the authors to address my original concerns with the first draft, and I think the revised version is improved. However, in my opinion there are several issues that should be addressed before I would consider the manuscript to be publishable. Importantly, the manuscript requires a careful read-through for typos as there are several throughout.

*We have revised in response to every comment from Reviewer 1, and all three authors have proof-read the revised text, hopefully catching all typos. We thank the reviewer for the careful reading and many constructive suggestions. Collectively, the comments and suggestions of both reviewers have led to a more rigorous presentation.*

L1; The title is not very specific. Suggest changing to something like "Benchmark seasonal prediction skill estimates based on regional indices".

*The title has been changed as suggested.*

L80-82; This statement is a bit trivial, and it's not clear what is being implied in relation to the previous sentence.

*This sentence has been deleted. The subsequent sentence has been made the start of a new paragraph.*

L87; Should state what the "perfect model" assumption is. Also, the perfect models don't just reveal information about persistence in the model (even though this can of course be computed from such simulations). Skill from perfect model simulations comes from a number of sources.

*The "perfect model" assumption is now stated in new text inserted in Lines 85-88. A new statement (Lines 92-94) makes the point that skill in "perfect models" is not attributable solely to persistence.*

L96; "Sea ice" → sea ice extent

*Corrected (now Line 99).*

L103; Should note that the first two references are based on statistical models, whereas the third is a conclusion from a dynamical model hindcast study and is only true for winter forecasts. It seems more appropriate to reference that sea ice thickness is an important source of predictive skill for the summer (references below), as this season is the main focus of this paper.

*Revised text (Lines 106-109) now distinguishes statistical and modeling studies in citations. Also, importance of ice thickness is noted in new sentences (Lines 109-113), where we have added the four suggested references and where we state explicitly that the present study does not include ice thickness.*

Day, J. J., E. Hawkins, and S. Tietsche, 2014b: Will Arctic sea ice thickness initialization improve seasonal forecast skill? Geophys. Res. Lett., 41, 7566–7575

Collow, T. W., W. Wang, A. Kumar, and J. Zhang, 2015: Improving Arctic sea ice prediction using PIOMAS initial sea ice thickness in a coupled ocean–atmosphere model. Mon. Wea. Rev., 143, 4618–4630

Dirkson, A., W. J. Merryfield, and A. Monahan, 2017: Impacts of sea ice thickness initialization on seasonal Arctic sea ice predictions. J. Climate, 30, 1001–1017

Zhang, Y., C.M. Bitz, J.L. Anderson, N. Collins, J. Hendricks, T. Hoar, K. Raeder, and F. Massonnet, 2018: Insights on Sea Ice Data Assimilation from Perfect Model Observing System Simulation Experiments. J. Climate, 31, 5911–5926

L118; While true, this study is only looking at how the trend impacts the anomaly correlation metric (a strictly mathematical consequence of that metric), whereas the Drobot 2003 study speculated that the statistical relationships between the predictors and the sea ice might change due to evolving physical relationships between them.

*We added the following statement (Lines 126-128): "…evolving physical relationships that underlie trend-related changes in statistical relationships are not addressed in the present study".*

L118-121; Not sure how this statement fits in with the rest of the paragraph, and the same point was already made in the preceding paragraph when referencing Bushuk et al 2017.

*Agreed, the text did not flow well here. We have moved the statement referencing Blanchard-Wrigglesworth et al. (2011) to the preceding paragraph (Lines 109-111).*

L149-152; Except that most of these studies used a long model control run to assess predictability, not restricted to the post 1979 satellite record.

*Added a statement (Lines 157-160) that the cited studies generally used long control runs from climate models, although their observational records were limited to the post-1979 period.*

L170; "Sea ice extent" --> "Pan-Arctic sea ice extent"

*"Pan-Arctic" inserted as suggested (Line 177).*

L176-180; could reference:
Lindsay, R., & Schweiger, A. (2015). Arctic sea ice thickness loss determined using subsurface, aircraft, and satellite observations. The Cryosphere, 9(1), 269-283.

*Reference added (Line 188).*

L179; CryoSat-2? CryoSat-1 wasn't successful. Also, reference needed.

*Deleted the mention of CryoSat (Lines 186-188).*

L261-266; Dirkson et al 2017 brought up this very issue, and suggested the use of a quadratic fit to detrend pan-Arctic sea ice area, differing the linear fit used by the methods cited here.

Dirkson, A., W.J. Merryfield, and A. Monahan, 2017: Impacts of Sea Ice Thickness Initialization on Seasonal Arctic Sea Ice Predictions. J. Climate, 30, 1001–1017

*Added statement (Lines 275-276) citing Dirkson et al. (2017), noting their suggested use of a quadratic fit.*

L292-295; Already said this in previous paragraph.

*Deleted the two sentences (Lines 302) that were repeated from preceding paragraph. More generally, in response to the reviewer's next few comments, we have rewritten the description of the curve-fitting procedure and its use in computing departures from the trend lines. The revised text is in Lines 295-320.*

L307-309; If `linregress' is the method used for fitting after finding the breakpoint year using ``curve_fit", this should be described in the methodology before describing Fig. 4. Don't you use this to detrend pan-Arctic SIE as well?

*Description of use of "linegress" has been revised for clarity (Lines 313-320), where we now state explicitly (Lines 324-315) that the two-piece linear fit was indeed used to detrend the time series.*

L319ff; But Fig 5 is based on non-detrended data, so not really comparable to Bushuk et al 2018.

*In order to enable a more direct comparison to Bushuk et al. (2018), we have added a second panel to Figure 5 showing the corresponding detrended results. The new panel (Fig. 5b) is described in a separate paragraph (Lines 344-357), where references to Bushuk at al. (2018) and other relevant studies are placed.*

L326-327; Much of the correlation here is due to the trend, is it not? A more informative comparison of the two periods would be based on detrended data.

*Correct. We have added a new panel (Fig. 5b) showing the corresponding results based on detrended data.*

L331; What statistical test is being used here?

*Added a new statement (Lines 339-340) that the significance testing was based on a Wald test with t-distribution of the test statistic and a two-sided p-value for a null hypothesis that the slope is zero.*

L337; Avoid using "significance" here if you're referring to the magnitude of the correlation and not statistical significance as measured according to a p-value.

*Changed the text (Lines 344-345) to say that "…the detrending generally reduces the magnitudes of the correlations…".*

L334; It would be more revealing to state these values in relation to the detrended values for the May and July auto-correlations. Again, this is why it would be useful to plot them on Fig 5.

*The new Figure 5b shows the values of the May and July auto-correlations. In the new text describing Fig. 5b (Lines 351-354), we discuss the contrast between the low and high correlations before and after the May "predictability barrier".*

L362ff; Which regions/months yield a breakpoint year in the 1960's? This feature appears in all months and should be mentioned and expanded on.

*In the new Lines 368-372, we list the regions/months with break-points in the 1960s, noting that there is no systematic pattern in space or time that would indicate a meaningful signal.*

*L369-370; This is redundant with the previous sentence.*

*Sentence has been deleted (Line 384).*

L373; Again, what test is being used?

*Added the following at the start of the sentence: "Based on the t-test described earlier,…".*
*See Lines 339-340 and response to reviewer's comment on L331 of previous manuscript.*

L378; correlations --> explained variance

*Corrected (Line 393).*

L385; why surprisingly?

*"somewhat surprisingly large" has been changed to "relatively large" in describing the*
*contribution of the Laptev Sea (Lines 399-401).*

L386-388; ...September extents <of> East Siberian…
<of> to <explained by>

*Lines 401-403 have been reworded as suggested.*

L397; "variations." to "variations in each region."

*Changed as suggested (Line 412).*

L411; five subregions. Didn't mention Laptev.

*Laptev Sea has been added to the list of five subregions (Lines 426-427).*

L428ff; The study referenced here is not a perfect-model study. Also, the comparison is indeed
not apples to apples, so I fail to see the point in making any comparison at all.

*Deleted the text (Lines 428-432 of previous manuscript) containing the comparison. The*
*paragraph now ends with the reference to Tables 1 and 2 (Line 443).*

L447; What region is being referred to here in terms of the explained variance in the BSI?

*Inserted explicit mention of the Beaufort, Chukchi and East Siberian Seas (Line 459).*

L454; Only in January-April is the trend in the Bering sea positive.

*Added a clarification (Lines 492-493) that the Bering sea ice trend is positive only in January-*
*March, citing Onarheim et al. (2018).*

L466; It would be more accurate to replace "predictive skill" throughout with "explained
variance". And whenever saying significant, please clarify if you are referring to statistical
significance.

*Changed as suggested (Line 504, Line 525 and elsewhere in Section 6).*

L471ff; low skill for pan-Arctic SIE based on what? Please be specific.

*Inserted "persistence-derived" as modifier of "skill for the detrended September pan-Arctic ice extent…" (Line 510).*

I find this statement conjectural. How do your results compare quantitatively with the SIO? And how can a comparison be drawn when that study is based on 7 years of data (a very short record) when correlation estimates aren't reported due to the small sample.

*Following the reviewer's subsequent suggestion (Line 484ff of previous manuscript), we have expanded the quantitative comparison of our results and the SIO (Lines 462-487). The sample of available SIO results is now 11 years. (vs. Stroeve et al.'s 7 years). While we do not present correlations, we do present RMS errors and median absolute errors of the SIO compared with the same metrics of skill of persistence forecasts made at the same lead times as the SIO. (See responses to the several of the following related comments by the reviewer).*

L474; shows that <the> spring.

*Inserted "the" (Line 513).*

L479; I find this sentence misleading and inaccurate. Several studies show forecast skill outperforming persistence using non-perfect model approach (i.e. initialized hindcasts) - including Bushuk et al 2018 which presents results using both approaches. There is a long list of these studies, and even if they do not make direct comparisons against persistence within, it is possible to compare their reported detrended correlations of SIE against the persistence values found here, at least in summary.

*This paragraph has been revamped (Lines 516-524) to distinguish the types of previous studies showing improvement over persistence. Four additional references are cited, including two new ones that have been added to the reference list: Lindsay et al. (2008) and Tivy et al. (2011).*

L480-482; This isn't very clear. The perfect model approach doesn't only reveal predictive skill from persistence; several sources of skill are included in these evaluations. Perfect model skill can be compared against persistence in the model, and initialized forecast skill can be compared against persistence in observations. Persistence timescales can also be compared between the model and observations, but I don't understand which comparison is being made here.

*Agreed. The statement on Lines 480-482 (previous manuscript) was confusing, and we have deleted it from the paragraph that now begins on Line 516.*

L484ff; Wouldn't it be more appropriate to compute these values based on anomaly persistence from June, July, and August SIE rather than year-to-year persistence? I.e. that which is presented in your "results" section, but for MAE and RMSE instead of explained variance or correlation. Are these values reported here based on all contributions to the SIO (heuristic, statistical, non-coupled models, fully-coupled models)? I'm not sure all of these would fall under "state of the art". Also, I think this analysis belongs in the results section rather than summary.

*As suggested, we have expanded this material and moved it to the section on Results; it is now Lines 462-487.  In the expanded comparison, we have included anomaly persistence statistics from May, June, July and August.  We also state explicitly that the SIO forecasts used in this comparison were averages of all forecasts submitted to SIO, so it is quite possible that some individual forecasters participating in the SIO perform considerably better.  Accordingly, we removed the phrase "state of the art" in connection with the SIO.*

L499-501; For which metrics? Based on explained variance from which metrics?

*Reworded Lines 525-527 to state that the metric is explained variance and to clarify what type of persistence (i.e., detrended anomaly persistence) is being referred to.*

Figs 5, 7-9; statistical significance should be incorporated into each of these figures.

*Since the significance thresholds are the same for all entries in a figure, we have added the significance thresholds to the captions of Figures 5, 7, 8 and 9.*

[revised manuscript text omitted]